

# Followership styles scrutinized: temporal consistency and relationships with job attitudes and self-efficacy

Mirko Ribbat[1], Christoph Nohe[2] and Joachim Hüffmeier[3]

[1] Federal Institute for Occupational Safety and Health, Dortmund, Germany
[2] University of Münster, Münster, Germany
[3] TU Dortmund University, Dortmund, Germany

## ABSTRACT

While followership has been repeatedly acknowledged as an important part of leadership, key questions are still awaiting empirical testing. In our two studies, we test Kelley's prominent concept of followership styles for the first time in a longitudinal design. Specifically, we use a latent-state trait approach to examine the degree to which followership behaviors (*i.e.*, active engagement [AE] and independent, critical thinking [ICT]) reflect rather stable or rather dynamic behaviors. Furthermore, we examine the relationships of followership behaviors with job attitudes (*i.e.*, job satisfaction and organizational commitment) and self-efficacy in latent states cross-lagged models. We first test our hypotheses in a sample of $N = 184$ employees from eleven German service organizations, which were surveyed twice with a time lag of nine to 12 months. To replicate and extend our findings from Study 1, we conducted Study 2 with a sample of $N = 570$ participants from a German open-access panel, which were surveyed twice with a time lag of four months. In Study 2, we additionally test leader humility and perceived organizational support (POS) as potential moderators of the relationships between followership and job attitudes. While our findings support Kelley's conceptualization of followership styles as rather consistent behavior patterns, mixed results were found for the relationships with the other variables. We discuss the theoretical and practical implications of our findings as well as the relevance of time in followership research.

## INTRODUCTION

The tradition of a classic leader-centric view (*i.e.,* the focus on the leader's role and characteristics) is still dominant in the field of leadership research and practice (*Avolio, Walumbwa & Weber, 2009*; *Banks et al., 2018*; *Dinh et al., 2014*). However, the specific roles and contributions of followers in the leadership process (*i.e.,* followership) have gained more attention in recent years (see, for instance, *Khan, Busari & Abdullah, 2019*; *Uhl-Bien et al., 2014*). In their integrative literature review and "Formal Theory of Followership" (FTF), *Uhl-Bien et al. (2014)* highlighted several ways to study how followers construe and enact their follower role, and how this may affect leaders and followership outcomes (*i.e.,* the role-based approach to followership; see *Uhl-Bien et al., 2014*). In addition,

Corresponding author
Mirko Ribbat,
mirko.ribbat@tu-dortmund.de

they introduced the constructionist perspective on followership, which centers on how individuals mutually interact to (co-)construct leadership and followership identities. Thus, according to the FTF (*Uhl-Bien et al., 2014*), followership either describes the role enactment from a predefined follower rank or position, or reflects an individual's actual act of following in a social interaction process. While the co-construction approach to followership is a rather novel approach within the field of followership research, the role-based approach has been developed over several decades (see, for instance, *Chaleff, 1995*; *Hurwitz & Hurwitz, 2015*; *Kelley, 1988*; *Kelley, 1992*).

Although *Kelley (1988)* provided one of the first (role-based) theories on the positive impact that followers can have in the leadership process, some of his assumptions are still awaiting empirical testing. Specifically, *Kelley (1992)* conceptualized different followership styles as rather stable behavior patterns based on the interaction of the followers' active engagement (AE) in the leadership process and their independent, critical thinking (ICT) toward their leader. The notion of stable followership behaviors was also echoed by subsequent approaches to followership styles, such as *Kelley (2008)*, *Khan, Busari & Abdullah (2019)*, or *Uhl-Bien et al. (2014)*. However, prior studies on Kelley's proposed followership behaviors[1] (*Blanchard et al., 2009*; *Gatti et al., 2014*; *Gatti, Ghislieri & Cortese, 2017*; *Ribbat, Krumm & Hüffmeier, 2021*) used cross-sectional designs and thus could not test his assumption of stable followership patterns. Indeed, there are various prominent approaches that consider behaviors at work as dynamic and variable (see, for instance, *Beal et al., 2005*; *Weiss & Cropanzano, 1996*). Therefore, followership behavior could also depend on situational factors such as followers' mood or the current environment in which they work (*Benson, Hardy & Eys, 2015*; *Weiss & Cropanzano, 1996*).

Furthermore, followership approaches stress the role of followers by considering followership as an important independent variable in leadership research instead of considering it as the dependent variable (*Uhl-Bien et al., 2014*; *Uhl-Bien & Carsten, 2018*). Thus, according to followership theory (*Kelley, 1988*; *Kelley, 1992*; *Uhl-Bien et al., 2014*), followership behavior should be a major predictor for followership outcomes (*e.g.*, individual follower outcomes such as job satisfaction or organizational commitment). On the individual follower level, prior research in fact demonstrated that AE and ICT correlate with job attitudes such as job satisfaction and organizational commitment (*Blanchard et al., 2009*; *Gatti et al., 2014*; *Ribbat, Krumm & Hüffmeier, 2021*). In contrast to followership approaches (*Uhl-Bien et al., 2014*; *Uhl-Bien & Carsten, 2018*), the literature on job attitudes and job performance (*e.g.*, *Riketta, 2008*), however, suggests that job attitudes precede follower behavior. Since prior research (*Blanchard et al., 2009*; *Gatti et al., 2014*; *Ribbat, Krumm & Hüffmeier, 2021*) cannot provide insights into the direction of relationships due to their cross-sectional designs, the direction of the relationship between followership and job attitudes is still unclear.

In the current research, we conducted two studies to test the temporal consistency of *Kelley*'s (*1992*) followership behaviors in a longitudinal design, along with their relations to critical job attitudes (*i.e.,* job satisfaction and organizational commitment) and self-efficacy for the first time. Specifically, we test both the construct stability of followership behaviors and the cross-lagged relationships between followership, job attitudes, and self-efficacy

[1] In the following, "followership behavior(s)" refer to *Kelley*'s (*1992*) definition and conception.

in two different samples. In Study 2, we additionally test two potential new moderator variables that were not part of Study 1 (*i.e.,* leader humility and perceived organizational support (POS)).

   Thus, our studies extend current research in several ways. First, we explore whether followership behavior can be characterized as rather stable or trait-like behavior patterns as proposed by *Kelley (1992)*. This is important because *Kelley*'s (*1992*) conception of styles suggests that followership should be conceptualized as rather trait-like behavioral tendencies that are related to rather stable personal characteristics and/or general circumstances. A state-like nature of followership behavior, however, would necessarily shift the focus of future research from general and typical factors to more specific, situational, and contingent factors as followership would then rather be spontaneous, dynamic, or variable. Second, we examine whether followership behaviors are antecedents of job attitudes (*i.e.,* job satisfaction and organizational commitment) as conceptualized by *Kelley (1992)* and/or vice versa in multistate models (*Geiser, 2020*; *Prenoveau, 2016*). These relationships have been investigated only in cross-sectional studies and, therefore, prior research could not yet provide a rigorous test of the direction of these relationships (see *Byun, Lee & Kang, 2018*; *Ribbat, Krumm & Hüffmeier, 2021*). Third, we explore the link of AE and ICT with self-efficacy, an important variable in the organizational context that has, however, not been studied thus far. According to *Kelley (1992)*, p. 143), active and critical followers are goal-oriented, success-oriented, and effective, which culminates in a "can do aura". Therefore, active and critical followership should be related to the followers' perception of self-efficacy. Finally, we consider leader humility and POS as moderators. Hence, we also explore two potential new conditions under which the relationships of followership with job attitudes might be fostered. In this way, we contribute to a better understanding of the followership construct (as conceptualized by *Kelley, 1992*) and potential followership outcomes.

## FOLLOWERSHIP

Followership behavior is defined as "behaviors of individuals acting in relation to a leader(s)" (*Carsten et al., 2010*, p. 545). In our study, we refer to the followership concept by *Kelley (1992)* that describes followership styles based on the interaction of the followers' active engagement (AE) in the leadership process and their independent, critical thinking (ICT) towards their leader. According to *Kelley (1992)*, the best followers are those who participate actively in the leadership process and take initiative. At the same time, they independently think for themselves and provide constructive criticism for their leader and group. By contrast, the worst followers do not independently think for themselves, simply take directions, and do not challenge their leader and group. Moreover, they are passive, lazy, and require constant supervision. *Kelley (1992)* proposes that the different combinations of AE and ICT result in five styles of followership behavior, which he describes as "passive" (*i.e.,* low in both dimensions), "conformist" (*i.e.,* high in AE, but low in ICT), "alienated" (*i.e.,* low in AE, but high in ICT), "exemplary" (*i.e.,* high in both dimensions), and "pragmatist" (*i.e.,* with medium levels in both dimensions). The

exemplary followership style is considered as most effective and most valuable to the organization (*Kelley, 1988*; *Kelley, 1992*).

In our current study, we will examine whether followership behavior can be characterized as rather stable behavior patterns as proposed by *Kelley (1992)*. To do so, we apply a latent state-trait approach (*Geiser, 2020*; *Steyer et al., 2015*). This approach can provide answers to the question of whether measurement instruments assess more trait-like or more state-like attributes. Specifically, a latent state-trait approach quantifies to which degree observed and/or underlying latent state ($\tau$) variables reflect trait effects that indicate consistency (*i.e.*, coefficient $\text{Con}_\tau$; see *Geiser, 2020*) or situation effects/person by situation interaction effects that indicate occasion specificity (*i.e.*, coefficient $\text{Osp}_\tau$; see *Geiser, 2020*). Measures can be considered as trait-like, when more than 50% of their true state variance is due to trait effects (*Geiser, 2020*; *Steyer, Schmitt & Eid, 1999*). In accordance with *Kelley*'s (*1992*) conceptualization of followership styles as rather stable (*i.e.,* trait-like) behavior patterns we predict:

*Hypothesis 1: (a) AE and (b) ICT will have a higher proportion of state variance at each time point that is due to trait effects (i.e., Con [ $\tau_{t1,t2}$ ] > .50) than state residual variance (i.e., Osp [ $\tau_{t1,t2}$ ] < .50).*[2]

[2]This study was preregistered on the Open Science Framework.

## Followership and job attitudes

Again, in accordance with Kelley's conceptualization, we posit that followership behavior will be related to important job attitudes (*i.e.*, job satisfaction and organizational commitment). Our Hypotheses are visualized in Figs. 1 and 2. Job satisfaction is defined as "a pleasurable or positive emotional state resulting from the appraisal of one's job or job experiences" (*Locke, 1976*, p. 1304). Organizational commitment "(a) characterizes the employee's relationship with the organization, and (b) has implications for the decision to continue or discontinue membership in the organization" (*Meyer & Allen, 1991*, p. 67). Empirical evidence suggests a positive relationship of AE with job satisfaction (*e.g.*, *Blanchard et al., 2009*; *Ribbat, Krumm & Hüffmeier, 2021*). With regard to ICT, several studies found no significant relationship of ICT with job satisfaction (*Gatti et al., 2014*; *Gatti, Ghislieri & Cortese, 2017*; *Ribbat, Krumm & Hüffmeier, 2021*). *Blanchard et al. (2009)* even reported that ICT was negatively associated with job satisfaction. Their results further suggest that the interaction of AE and ICT increases intrinsic job satisfaction (*i.e.,* satisfaction with job aspects that are task-related such as responsibility) but decreases extrinsic job satisfaction (*i.e.,* satisfaction with job aspects that are unrelated to the task such as the supervisor; *Blanchard et al., 2009*). Specifically, followers with high AE and high ICT had the highest levels of intrinsic job satisfaction. The more followers engaged in ICT, the lower was their extrinsic job satisfaction, while this effect was weaker when followers were also actively engaged (*Blanchard et al., 2009*). The findings of *Favara (2009)* and *Saraih et al. (2018)* also showed a positive relation of an exemplary followership style (*i.e.,* high scores both on AE and ICT) with job satisfaction. Furthermore, two studies (*Blanchard et al., 2009*; *Ribbat, Krumm & Hüffmeier, 2021*) showed that AE was positively associated with organizational commitment, whereas ICT was negatively related to organizational commitment.

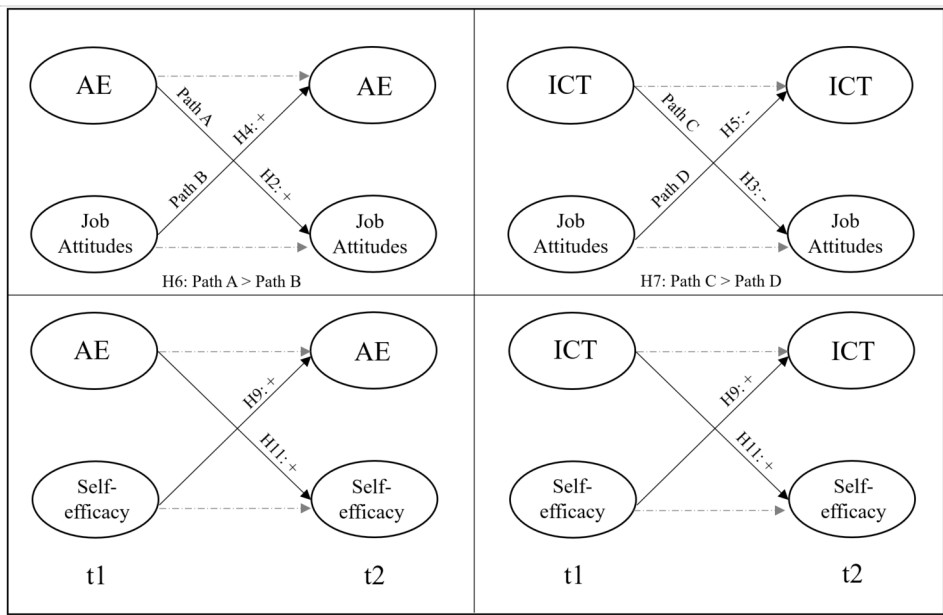

**Figure 1  Hypotheses for the cross-lagged models.**

To derive our first set of hypotheses we apply Cognitive Dissonance Theory (*Festinger, 1957*; *Harmon-Jones & Mills, 2019*). According to Cognitive Dissonance Theory, employees tend to harmonize conflicting cognitions (*e.g.*, attitudes and behaviors). We follow the effort justification tenet of Cognitive Dissonance Theory (*Aronson & Mills, 1959*; *Harmon-Jones, Armstrong & Olson, 2018*), which refers to the strategy of adding consonant cognitions to dissolve experienced dissonance when undertaking demanding or exhausting activities and achieving outcomes that may not be satisfying. Thus, we expect followers to value their job and their organization more to justify the considerable effort they spend on engaging actively in the leadership process (see also *Blanchard et al., 2009*; *Ribbat, Krumm & Hüffmeier, 2021*). Hence, their earlier active engagement (at t1) is expected to increase their later job attitudes (at t2). This reasoning is in line with the dominant view that AE antecedes followers' job attitudes (*Blanchard et al., 2009*; *Gatti et al., 2014*; *Ribbat, Krumm & Hüffmeier, 2021*). Thus, we predict:

*Hypothesis 2a: AE at t1 will be positively related to job satisfaction at t2.*

*Hypothesis 2b: AE at t1 will be positively related to organizational commitment at t2.*

However, we expect that ICT at t1 exerts a negative influence on attitudinal variables at t2, because it increases the followers' awareness of the problems and negative aspects of their job. Hence, rather than valuing their job and organization more due to the effort that has also to be expended on independent, critical thinking, followers with high levels of ICT should become less satisfied and committed when faced with the problems and negative aspects of their job (see also *Blanchard et al., 2009*; *Ribbat, Krumm & Hüffmeier, 2021*). Therefore, we hypothesize:

*Hypothesis 3a: ICT at t1 will be negatively related to job satisfaction at t2.*

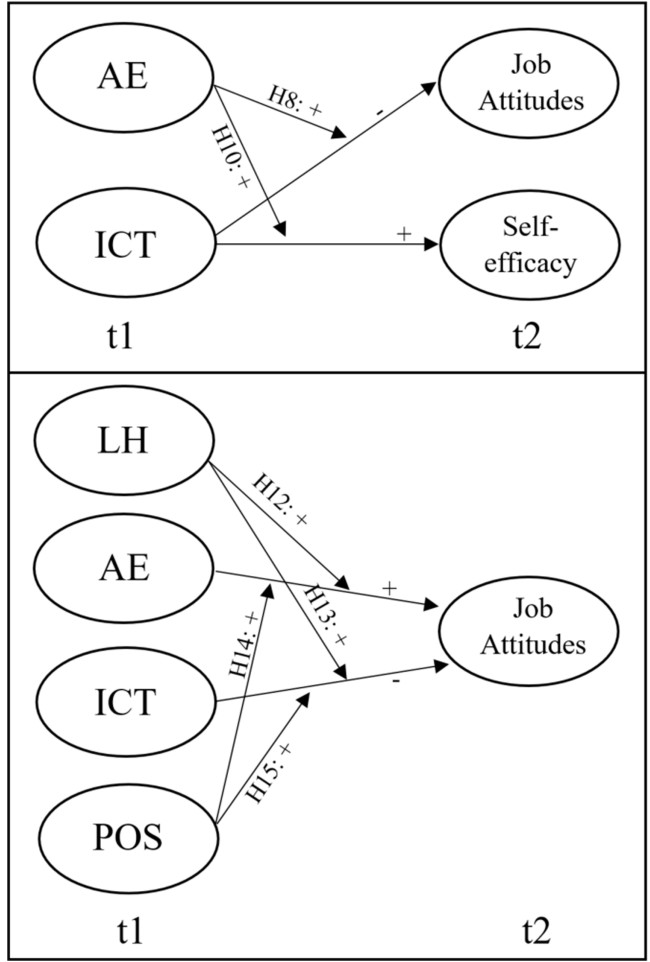

**Figure 2** Hypotheses for the interaction effects.

*Hypothesis 3b: ICT at t1 will be negatively related to organizational commitment at t2.*

Besides effort justification, another strategy to avoid cognitive dissonance according to Cognitive Dissonance Theory (*Festinger, 1957*; *Harmon-Jones & Mills, 2019*) would be to bring behaviors in line with one's attitudes. That is, attitudes can guide and facilitate behavior. This notion suggests that job attitudes can cause later job behaviors (*Hinojosa et al., 2017*; *Riketta, 2008*). Therefore, we expect satisfied and committed followers to actively engage in the leadership process. We state:

*Hypothesis 4a: Job satisfaction at t1 will be positively related to AE at t2.*

*Hypothesis 4b: Organizational commitment at t1 will be positively related to AE at t2.*

However, we do not expect satisfied and committed followers to question the leader and the organization frequently. Since satisfaction and commitment are pleasurable and positive emotional states (*Locke, 1976*; *Meyer & Allen, 1991*), there should be no incentive for the followers to create conflicting cognitions and risk psychological discomfort (*Festinger, 1957*; *Harmon-Jones & Mills, 2019*). Thus, satisfied and committed followers might avoid

becoming aware of negative aspects of their job through ICT in order to prevent cognitive dissonance. Hence, we predict that job attitudes at t1 will be negatively related to ICT at t2.

*Hypothesis 5a: Job satisfaction at t1 will be negatively related to ICT at t2.*

*Hypothesis 5b: Organizational commitment at t1 will be negatively related to ICT at t2.*

Additionally, we argue that the effort justification mechanism (cf. Hypothesis 2) is stronger than the "attitudes as guidelines" mechanism (cf. Hypotheses 4 and 5), because followership styles, which are conceptualized as relatively stable or trait-like behavior patterns (*Kelley, 1992*; see also our Hypothesis 1), should be more difficult to change than job attitudes. Therefore, we expect stronger relations of AE and ICT at t1 with the attitudinal variables at t2 due to effort justification than the opposite relations between attitudes (at t1) and AE and ICT (at t2). Thus, we hypothesize:

*Hypothesis 6: The relationships of AE at t1 with (a) job satisfaction at t2 and (b) organizational commitment at t2 is stronger as compared to the relationships of (a) job satisfaction at t1 and (b) organizational commitment at t1 with AE at t2.*

*Hypothesis 7: The relationships of ICT at t1 with (a) job satisfaction at t2 and (b) organizational commitment at t2 is stronger as compared to the relationships of (a) job satisfaction at t1 and (b) organizational commitment at t1 with ICT at t2.*

Since *Kelley*'s (*1992*) typology of followership styles refers to the interaction of both followership dimensions (*i.e.,* AE and ICT), we also investigate this interaction. Continuing the argumentation above, the less followers engage actively in the leadership process, the more should the negative influence of ICT on job attitudes prevail over effort justification. However, independent, critical followers might also use active engagement to change undesired working conditions and, thus, they should become more satisfied and committed. Therefore, following the logic of both Cognitive Dissonance Theory (*Festinger, 1957*) and *Kelley*'s (*1992*) conceptualization, we predict:

*Hypothesis 8: ICT at t1 will be less negatively related to (a) job satisfaction at t2 and (b) organizational commitment at t2 when AE at t1 is high as compared to when AE at t1 is low.*

## FOLLOWERSHIP AND SELF-EFFICACY

Self-efficacy is defined as "people's judgements of their capabilities to organize and execute courses of action required to attain designated types of performances" (*Bandura, 1986*, p. 391). According to *Kelley (1992)*, active and critical followers are goal-oriented, success-oriented, and effective. He describes effective followers as those who determine early and accurately the criteria for success in the leadership environment, who track a record of successes in tasks that are important to the leader, and who "seize smaller victories that accumulate into a 'can do' aura" (*Kelley, 1992*, p.143). Therefore, active and critical followership should raise the followers' awareness for their own capabilities to attain personal and leadership goals. Consequently, active and critical followership should be related to the followers' perception of self-efficacy. Prior meta-analyses (*Sadri & Robertson, 1993*; *Stajkovic & Luthans, 1998*) demonstrated that self-efficacy is positively related to work-related behaviors (*e.g.*, changes in career tracks or intentions to show assertiveness) and job performance. However, the relation of this important individual-level construct

with followership behavior has not been tested so far. Therefore, we will investigate this relationship in our study.

According to Social Cognitive Theory (*Bandura, 1997*), self-efficacy emerges from four sources: (1) physiological and affective states, (2) verbal persuasion, (3) vicarious experience, and (4) mastery experience. Mastery experience is considered to have the strongest impact on the development of self-efficacy, because it provides authentic evidence for one's capability to succeed, and, thus, builds a robust belief in one's personal efficacy (*Bandura, 1997*). When followers simply take directions, there is no evidence for the followers' own capability to succeed, because success fully depends on the leader's decisions. Active and critical followership, however, allow for evaluating the followers' own contributions to leadership success (*e.g.*, followers' high individual or collective job performance) and, thus, raises the awareness for mastery experiences. Therefore, we expect that the followers' self-efficacy substantially results from ascribing (at least parts of) the leadership success to own efforts when carrying out an active and independent follower role. Hence, we expect that both AE at t1 and ICT at t1 are positively associated with self-efficacy at t2, which is also in line with the dominant view of followership research (*Kelley, 1992*; *Uhl-Bien et al., 2014*) that followership should be a major predictor for followership outcomes:

*Hypothesis 9a: AE at t1 will be positively related to self-efficacy at t2.*

*Hypothesis 9b: ICT at t1 will be positively related to self-efficacy at t2.*

Paralleling our above hypotheses on job attitudes, the relationship of ICT (at t1) with self-efficacy (at t2) might depend on the level of AE. While it may be obvious to ascribe leadership success to own efforts (*i.e.*, to allow for mastery experience; see *Bandura, 1997*) when active engagement is high, the contribution of ICT to leadership success might not always be apparent. This should, for instance, be the case when a follower expresses concerns about the leader's direction and consequently does not (strongly) contribute to goal attainment. According to *Kelley (1992)*, followers may become frustrated and "alienated", when their concerns are ignored repeatedly, which keeps them away from their goal achievements and day-to-day job satisfaction (see *Kelley, 1992*). When active engagement is high, however, ICT could be considered as an additional contribution to leadership success and therefore raise even more awareness for mastery experiences than the active engagement alone. Thus, we predict:

*Hypothesis 10: ICT at t1 will be more positively related to self-efficacy at t2 when AE at t1 is high than when AE at t1 is low.*

Furthermore, according to Social Cognitive Theory (*Bandura, 1997*), self-efficacy is also considered to be a predictor for one's choices of activities, effort expenditure, persistence, thought patterns, and emotional reactions when confronted with obstacles (*Lent, Brown & Hackett, 1994*). Followers with high self-efficacy should therefore be more likely to actively engage in the leadership process and rather be willing to act independently and critically. Thus, we predict:

*Hypothesis 11a: Self-efficacy at t1 will be positively associated with AE at t2.*

*Hypothesis 11b: Self-efficacy at t1 will be positively associated with ICT at t2.*

[3]Our research is in line with the ethical principles of the Declaration of Helsinki and the Psychologists Code of Conduct of the American Psychological Association. Please note that correlational studies are exempt from institutional review in Germany (see also *Sonnentag et al., 2022*). As prescribed by the German Medicines Laws (AMG, MPDG) and the associated EU regulations (CTR 536/2014, MDR 2017/745), institutional ethical approval is mandatory if research involves drug or medical device testing, or if physicians participate. Our studies do not meet any of those criteria. However, we obtained retrospective ethical approval of the University of Münster for Study 2 (approval number 2022-65-ChN). For Study 1, the data had been collected by the German Federal Institute for Occupational Safety and Health in a very similar way as Study 2.

[4]One reason for the relatively high drop-out could be that we had to rely on third persons (*i.e.*, organizational multipliers) to recruit the respondents within the organizations. Furthermore, we did not pay participants for each response. Another reason could be that the matching process was realized *via* a self-chosen code. The need for retaining the self-chosen code over nine to 12 months might have resulted in mistakes in some cases. Hence, there might be more respondents that completed both questionnaires, which, however, could not be matched *via* the personal code. However, the resulting sample of participants for which the questionnaire could be matched for both time points is comparable to the original sample at time point one in terms of their mean age, gender distribution, level of education, and professional sector distribution. Furthermore, the drop-out rate is comparable to studies with a similar approach to data collection (see, for instance, *Rayton & Yalabik, 2014*; *Tims, Bakker & Derks, 2015*).

# STUDY 1

## Materials and methods

This study was preregistered at the Open Science Framework (https://osf.io/tf493/?view_only=03c75588fa514565be62adbdc58b24dc). Following *Simmons, Nelson & Simonsohn (2012)*, we report and explain in detail (a) how we determined our sample size, (b) all exclusions, and (c) all independent and dependent variables (see also *Simmons, Nelson & Simonsohn, 2012*).

## Sample

The variables that we investigated in this study were part of a more comprehensive data set, which had been collected for the German Federal Institute for Occupational Safety and Health between 2017 and 2018 (https://www.baua.de/EN/Tasks/Research/Research-projects/f2372.html) in German language. Employees and their supervisors from eleven German service organizations were surveyed twice with an online questionnaire. Participants gave their consent for participation within the questionnaire by checking the associated box.[3] In this study, we only used data from the employees' surveys. The time lag between the two measurement waves (*i.e.*, t1 and t2) was between nine and 12 months. Persons responsible for human resources in the participating organizations invited various teams to complete the survey *via* e-mail. Therefore, we have no information about the exact number of employees that were invited to the survey. The first measurement wave was answered by 551 employee responses. For the second wave, employees who were originally invited had again the opportunity to participate, regardless of whether they had participated in the first wave. The second measurement wave was answered by 349 employees in total. For 187 employees (or 34% of the original population), we could match the questionnaires of both time points.[4]

Following our preregistration, some participants were excluded from the data set because of responding in a careless manner. Following the procedures recommended by *Meade & Craig (2012)*, three outlier cases were identified by computing the Mahalanobis Distance over all items, which reduced the overall sample size to 184 employees. Moreover, for five persons, the followership values were recoded as missing values either for t1 or t2 due to zero-within-variance in responses, resulting in a sample of 179 participants. The respondents were mainly employed within the public service sector (76%). Another 17% worked within the finance service sector (banking or insurance). A small proportion (7%) was employed in other service organizations (*i.e.*, health services or information technology services). The mean age was 43.5 years and 67% of the respondents were female, 33% were male. The level of education was distributed as follows: 43% had a completed apprenticeship, 40% had a university of applied science degree, and 17% had a university degree.

## Measures

Both dimensions of followership behavior (*i.e.*, AE and ICT) were measured using the German version of *Kelley*'s (*1992*) Followership Questionnaire by *Ribbat, Krumm & Hüffmeier (2021)*; with 10 items for AE and four items for ICT. An exemplary item for AE

was "Do you understand the leader's needs, goals, and constraints, and work hard to help meet them?". The ICT subscale, for instance, included the item "Do you assert your views on important issues, even though it might mean conflict with your group or reprisals from the leader?". Possible responses ranged from *1 ([almost] never)* to *7 ([almost] always)*.

Job satisfaction was measured with six items of the "Copenhagen Psychosocial Questionnaire" (*Nübling et al., 2005*). The respondents were asked to rate their satisfaction with their colleagues, leadership, challenges of work, use of abilities, career perspective, and job satisfaction overall. The response options ranged from *very dissatisfied (1)* to *very satisfied (7)*.

Organizational commitment was measured with three items of the scale from *Mowday, Steers & Porter (1979)* in the German Version by *Maier & Woschée (2002)*. A sample item was "I talk up this organization to my friends as a great organization to work for". Response options ranged from *strongly disagree (1)* to *fully agree (7)*. Finally, we measured self-efficacy using the scale by *Rigotti, Schyns & Mohr (2008)* with six items. A sample item was "I feel up to most of the job demands". Response options ranged from *not correct at all (1)* to *applies completely (7)*.

## Data analysis strategy

We applied a latent state-trait approach (*Geiser, 2020*; *Steyer et al., 2015*). This approach allows for differentiating trait effects from situation/person situation interaction effects (*Steyer, Schmitt & Eid, 1999*; *Steyer et al., 2015*). Specifically, we applied single-trait multi-state models with indicator-specific residual factors (STMS-IS; see *Geiser, 2020*) for both followership dimensions (*i.e.,* AE and ICT). The single-trait multi-state model (STMS) refers to the test of a single trait (*i.e.,* AE or ICT) in multiple situations (*i.e.,* t1 and t2). The indicator-specific residual factors account for indicator specific effects of latent variables (*Eid, Schneider & Schwenkmezger, 1999*; *Geiser, 2020*). This is important, because the simple STMS model would assume perfect homogeneity of the indicators for the latent variable that are measured at the same time point (*Geiser, 2020*). However, this was not a realistic assumption, as both AE and ICT measures were not perfectly consistent (see Table 1). Thus, the STMS-IS model was more appropriate to test the consistency of the followership measures, as it reflects potential method effects of indicator heterogeneity. The model fit was evaluated using the resulting chi-square values, the Root Mean Square Error of Approximation (RMSEA), the Comparative Fit Index (CFI), and the Standardized Root Mean Square Residual (SRMR). The following cut-off values were considered to indicate a good model fit (*West, Taylor & Wu, 2012*): $\chi^2/df < 5$, RMSEA $< .06$, CFI $> .95$, and SRMR $< .08$.

In order to test the hypotheses on the interrelations of followership with the other study variables, we applied latent autoregressive/cross-lagged states models (LACS; see *Geiser, 2020*). Latent states cross-lagged models are longitudinal models that allow for testing relationships of latent state variables over multiple states (*i.e.,* t1 and t2) in both directions in the same model, while correcting for random measurement error. By controlling for the autoregressive effects of the variables (*i.e.,* the autoprediction of the dependent variables), LACS models additionally take into account that in social science previous states are

**Table 1 Descriptives, intercorrelations between latent variables, and internal consistencies of Study 1.**

| Model | M | SD | 1 | 2 | 3 | 4 | 5 | 6 | 7 | 8 | 9 | 10 |
|---|---|---|---|---|---|---|---|---|---|---|---|---|
| 1. Followership: AE (t1) | 5.18 | 0.87 | .96″ | | | | | | | | | |
| 2. Followership: ICT (t1) | 4.62 | 1.00 | .41*** | .75″ | | | | | | | | |
| 3. Job satisfaction (t1) | 5.03 | 1.14 | .20* | −.11 | .86′ | | | | | | | |
| 4. Organizational commitment (t1) | 5.34 | 1.18 | .27** | −.05 | .56*** | .78′ | | | | | | |
| 5. Self-efficacy (t1) | 5.27 | 0.95 | .58*** | .31*** | .20* | .31*** | .87′ | | | | | |
| 6. Followership: AE (t2) | 5.16 | 0.88 | .80*** | .24* | .33*** | .32*** | .52*** | .98″ | | | | |
| 7. Followership: ICT (t2) | 4.79 | 0.91 | .32** | .86*** | .08 | .16 | .36*** | .41*** | .69″ | | | |
| 8. Job satisfaction (t2) | 5.96 | 1.12 | .05 | .01 | .72*** | .41*** | .12 | .17* | .04 | .88′ | | |
| 9. Organizational commitment (t2) | 5.20 | 1.25 | .27** | .03 | .45*** | .96*** | .34*** | .32*** | .26* | .39*** | .75′ | |
| 10. Self-efficacy (t2) | 5.27 | 0.89 | .45*** | .18 | .16* | .31*** | .81*** | .58*** | .34*** | .16 | .31*** | .86′ |

Notes.
  $N = 184$.
  AE, Active engagement; ICT, Independent, critical thinking.
  Values along the diagonal represent internal consistency ($'\omega$ or $''\omega$s).
  *$p < .05$.
  **$p < .01$.
  ***$p < .001$.

usually strong predictors for future states of the same variable (*Adachi & Willoughby, 2015*; *Geiser, 2020*). We also considered potential method effects of indicator heterogeneity and applied models with indicator-specific residual factors (Eid et al., 1999; *Geiser, 2020*). For the models that tested the interaction effects of AE and ICT, we also controlled for the autoregressive effects of job attitudes and self-efficacy, and applied indicator-specific residual factors. We used one-tailed tests of significance for the regression coefficients, since we had directed hypotheses (*Cho & Abe, 2013*; *Jones, 1952*; *Lakens, 2016*).

As our measures of internal consistency, we used a reliability indicator based on factor models and report coefficient omega ($\omega$; *McDonald, 1978*). We report the omega subscale ($\omega_s$) for multidimensional constructs (*i.e.*, for each followership dimension) as described by *Rodriguez, Reise & Haviland (2016)*. We used *Watkins (2013)* standalone program to compute the omega subscale ($\omega_s$). Every other analysis was computed with MPlus 7.4 (*Muthén & Muthén, 2015*).

### Pre-analyses

*Construct validity.* In our study, we analyzed four latent variables including five latent factors at two measurement times: followership behavior with its two dimensions AE and ICT, job satisfaction, organizational commitment, and self-efficacy. In order to ensure the distinctiveness of our study variables, we compared our five-factor measurement model to an alternative four-factor model that specified both job attitudes (*i.e.*, job satisfaction and organizational commitment) as one common factor. Chi square difference tests revealed that our measurement model fitted the data better than the alternative model both at t1, $\chi^2_{\mathrm{diff}}(3) = 96.43$, $p < .001$, and at t2, $\chi^2_{\mathrm{diff}}(3) = 111.98$, $p < .001$. Hence, our analyses indicate the distinctiveness of our study variables.

Note, however, that we allowed the residual correlation between the two items "Instead of waiting for or merely accepting what the leader tells you, do you personally identify

which organizational activities are most critical for achieving the organization's priority goals?" and "Do you actively develop a distinctive competence in those critical activities so that you become more valuable to the leader and the organization?" in the followership model ($r = .71$, $p < .001$ for t1; $r = .52$, $p < .001$ for t2). In their validation study for the German version of *Kelley*'s (*1992*) followership questionnaire, *Ribbat, Krumm & Hüffmeier (2021)* argued for this model specification as the latter item was formulated with a direct reference to the former. In addition, the two concerned items reflect the efforts to achieve overarching organizational goals and thus share a plausible commonality (*Ribbat, Krumm & Hüffmeier, 2021*). In our study, allowing the respective error term correlation led to better model fit both at t1, $\chi^2_{\text{diff}}(1) = 119.43$, $p < .001$, and at t2, $\chi^2_{\text{diff}}(1) = 46.47$, $p < .001$.

*Measurement equivalence across time.*  Before testing the hypotheses, we tested for measurement equivalence across time to follow a common standard for our analyses: "ME is a prerequisite for meaningful across-time comparisons […] Without measurement equivalence, differences in latent state factor means or variances across occasions may be due to changes in measurement (observed variable) properties rather than true changes in latent variables" (*Geiser, 2020*, pp. 122–123). Following *Geiser (2020)*, we consecutively tested measurement equivalence models that differ by the level of measurement equivalence (*i.e.,* various parameter equality constraints) for every latent variable. First, we tested for configural invariance that specified the same number of factors and the same factor loading pattern across time. Second, we also constrained the factor loadings to remain the same for a given observed variable in addition to configural invariance (weak invariance). Subsequently, we tested for strong invariance (strong measurement equivalence), which additionally set the intercepts to remain the same across time for a given observed variable. Finally, the strict invariance model (strict measurement equivalence) additionally determined the measurement error variance to remain the same across time for a given variable. Researchers typically aim for at least strong measurement equivalence, because it allows for meaningful comparisons of latent variable means and variances across time (*Meredith, 1993*; *Widaman & Grimm, 2014*).

The fit of the various measurement equivalence models is presented in the Supplemental Material S1 (SM1). In our model comparisons, strong measurement equivalence was preferred for AE and strict measurement equivalence was preferred for ICT. Thus, for AE and ICT at least strong measurement equivalence could be assumed. This was also the case for the other latent variables: Strong measurement equivalence was preferred for organizational commitment and self-efficacy. For job satisfaction, the strict model was the best fitting model. Hence, we used these models in further analyses.

## RESULTS

Table 1 shows the descriptive statistics, the intercorrelations of the latent variables and the reliabilities of the measures. For ICT at t2, $\omega_s = .69$ was slightly below the commonly used minimum value (*Dunn, Baguley & Brunsden, 2014*; *Kline, 1998*). However, we addressed this problem by specifying latent state-trait models with indicator-specific residuals (*Eid, Schneider & Schwenkmezger, 1999*; *Geiser, 2020*; see also above).

### Temporal consistency of followership behavior

To test Hypothesis 1, we used a STMS-IS model, which fitted the data well, $\chi^2(df) = 207.53(141)$, $p < .001$, $\chi2/df = 1.47$, RMSEA $= .05$, CFI $= .96$, SRMR $= .06$. The results revealed that AE was rather stable and trait-like over time as indicated by consistencies exceeding the commonly applied 50% threshold for both AE at time 1 (Con [ $\tau_{AE,t1}$] $= .78$, $p < .001$,) and AE at time 2 (Con[ $\tau_{AE,t2}$] $= .74$, $p < .001$) as compared to the coefficients that indicate occasion specificity (Osp[ $\tau_{AE,t1}$] $= .22$, $p = .001$, Osp[ $\tau_{AE,t2}$] $= .26$, $p < .001$). Hence, our results suggest that AE was consistent across time and rather trait-like, which supported Hypothesis 1a.

The results of the STMS-IS model for ICT revealed Con[ $\tau_{ICT,t1}$] $= .59$, $p < .001$, Con[ $\tau_{ICT,t2}$] $= .82$, $p < .001$, Osp[ $\tau_{ICT,t1}$] $= .41$, $p < .001$, and Osp[ $\tau_{ICT,t2}$] $= .18$, $p < .001$. The model fit was good: $\chi^2(df) = 35.89(24)$, $p = .06$, $\chi2/df = 1.50$, RMSEA $= .05$, CFI $= .96$, SRMR $= .08$. Thus, ICT, was consistent across time and rather trait like. Hence, Hypothesis 1b was also supported.

### Followership and job attitudes

The first cross-lagged model explored the relationship of AE with job satisfaction. Both autoregressive paths were significant (for AE: $\beta = .75$, $p < .001$; for job satisfaction: $\beta = .71$, $p < .001$). The relationship of AE (at t1) with job satisfaction (at t2) was not significant, $\beta = -.08$, $p = .12$, thereby not supporting Hypothesis 2. However, job satisfaction (at t1) was positively related to AE (at t2), $\beta = .13$, $p = .02$, thereby supporting Hypothesis 4a. The model explained 60% of the variance in AE and 49% of the variance in job satisfaction and the model fit was good: $\chi^2(df) = 524.84(371)$, $p < .001$, $\chi2/df = 1.42$, RMSEA $= .05$, CFI $= .95$, SRMR $= .06$.

The second cross-lagged model included AE and organizational commitment. Both autoregressive paths were significant (for AE: $\beta = .74$, $p < .001$; for organizational commitment: $\beta = .87$, $p < .001$). The relationship of AE (at t1) with organizational commitment (at t2) was not significant, $\beta = .02$, $p = .39$, which does not support Hypothesis 2. However, organizational commitment (at t1) was positively related to AE (at t2), $\beta = .11$, $p = .049$, thereby supporting Hypothesis 4b.[5] The model explained 59% of the variance in AE and 75% of the variance in organizational commitment and the model fit was good: $\chi^2(df) = 325.76(236)$, $p < .001$, $\chi2/df = 1.38$, RMSEA $= .05$, CFI $= .96$, SRMR $= .05$.

The relationship of ICT with job satisfaction was explored in the third cross-lagged model. While we had predicted this relationship to be negative, our results rather pointed in a positive direction. Thus, we applied a two-tailed test for this model. We, therefore,

[5] Please note that this relationship was no longer significant when we additionally controlled for age and gender. Controlling for age and gender did not, however, significantly affect any other tested relationship.

could neither find a significant relation of job satisfaction (at t1) to ICT (at t2), $\beta = .16$, $p = .08$, nor between ICT (at t1) and job satisfaction (at t2), $\beta = .09$, $p = .21$. These results do not support Hypotheses 3a and 5a. Both autoregressive paths were significant (for ICT: $\beta = .74$, $p < .001$; for job satisfaction: $\beta = .71$, $p < .001$). The model explained 53% of the variance in ICT and 50% of the variance in job satisfaction and the model fit was again good: $\chi^2(df) = 201.37(146)$, $p = .002$, $\chi 2/df = 1.38$, RMSEA $= .05$, CFI $= .97$, SRMR $= .06$.

The fourth cross-lagged model explored the relationship of ICT with organizational commitment. Again, we applied a two-tailed test for the model, because our results pointed in another direction than we had expected. Both autoregressive paths were significant (for ICT: $\beta = .72$, $p < .001$; for organizational commitment: $\beta = .86$, $p < .001$). No significant relationship of ICT (at t1) with organizational commitment (at t2) could be found, $\beta = .06$, $p = .41$. Organizational commitment (at t1) was also not related to ICT (at t2), $\beta = .17$, $p = .07$. These results do not support Hypotheses 3b and 5b. The model explained 55% of the variance in ICT and 75% of the variance in organizational commitment and the model fit was good: $\chi^2(df) = 80.71(65)$, $p = .09$, $\chi 2/df = 1.24$, RMSEA $= .04$, CFI $= .98$, SRMR $= .06$.

In Hypotheses 6 and 7, we proposed the relationships of AE (at t1) and ICT (at t1) with job attitudes (at 2) to be stronger than those of job attitudes (at t1) with AE (at t2) and ICT (at t2). Both hypotheses were not supported by our results, because we found no significant relationship of AE or ICT (at t1) with any job attitude (at t2). The relationship of AE (at t1) with job satisfaction (at t2) was not significant, $\beta = -.08$, $p = .12$. The relationship of AE (at t1) with organizational commitment (at t2) was not significant, $\beta = .02$, $p = .39$. The relationship between ICT (at t1) and job satisfaction (at t2) was also not significant, $\beta = .09$, $p = .21$. Finally, no significant relationship of ICT (at t1) with organizational commitment (at t2) could be found, $\beta = .06$, $p = .41$.

Furthermore, in Hypothesis 8 we posited ICT (at t1) to be less negatively related to job satisfaction and organizational commitment (at t2) when AE (at t1) was high, as compared to when AE (at t1) was low. In our analysis, the interaction of AE (at t1) and ICT (at t1) was not a significant predictor for job satisfaction (at t2), $\beta = -.10$, $p = .14$. However, the interaction (at t1) predicted organizational commitment (at t2), $\beta = .14$, $p = .04$, while controlling for the autoregressive effect of organizational commitment (at t1), $\beta = .86$, $p < .001$. The interaction effect is plotted in Fig. 3. Overall, the model explained 78% of the variance. Since the interaction of AE and ICT (at t1) was associated with a higher score in organizational commitment, Hypothesis 8b was supported.

## Followership and self-efficacy

We explored the relationship of AE and ICT with self-efficacy in different cross-lagged models. In addition, we tested whether the interaction of AE (at t1) and ICT (at t1) could predict self-efficacy (at t2). We could not find a significant relationship of AE (at t1) with self-efficacy (at t2), $\beta = -.02$, $p = .37$, nor did ICT (at t1) predict self-efficacy (at t2), $\beta = -.09$, $p = .11$. Hence, Hypothesis 9 was not supported. The interaction of AE (at t1) and ICT (at t1) did also not predict self-efficacy (at t2), $\beta = .01$, $p = .47$. Thus, Hypothesis

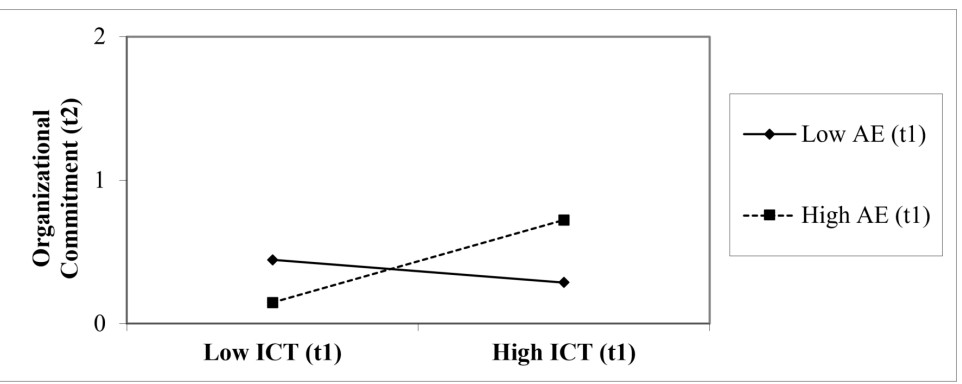

**Figure 3** **Interaction Effect of AE (t1) and ICT (t1) on Organizational Commitment (t2).** *Note.* AE, Active Engagement; ICT, Independent, Critical Thinking.

10 was not supported. Finally, we found no evidence for Hypothesis 11, as no significant relationship of self-efficacy (at t1) and AE (at t2) was detected, $\beta = .07$, $p = .15$, and as self-efficacy (at t1) did also not predict ICT (at t2), $\beta = .08$, $p = .19$.

## BRIEF DISCUSSION

Overall, we found evidence for some, but not for all of our hypotheses. The results of Study 1 support *Kelley*'s (*1992*) assumption that followership behaviors are rather stable behavior patterns. Since we found no evidence for the relationships of AE or ICT (at t1) and later job attitudes or self-efficacy (at t2), our results of Study 1 challenge the original idea of followership theory, that followership behaviors are significant predictors for important organizational variables (see, for instance, *Kelley, 1992*; *Uhl-Bien et al., 2014*). However, the interaction of AE and ICT (at t1) was positively related to organizational commitment (at t2), which corresponds to *Kelley*'s (*1992*) assumption that exemplary followership leads to desired organizational variables. One reason for the absence of the expected effects could be the length of the time lag between the two measurement waves (nine to 12 months). That is, some effects might have simply vanished over time and, thus, might not have been possible to detect. This would be the case, for instance, if the respondents' leader changed between the two measurement waves, if the proposed effects unfolded too quickly, or if the effects did not last long enough to be detected (for a related analysis of time-lags in leadership research, see *Fischer, Dietz & Antonakis, 2017*).

Another reason could be that we did not include conditions under which the relationships of followership with job attitudes might be fostered in our first study. *Kelley (1988)* argued that active and critical followers are the most effective and, thus, active and critical behaviors should be positively related to followership outcomes. However, he also admitted that an exemplary followership style might not always be the preferred style of a certain leader or organization. Therefore, exemplary followership might not always be acknowledged or rewarded. The missing acknowledgement of the followers' efforts of engaging actively

and critically in the leadership process could explain why followership behavior does not necessarily increase the follower's job satisfaction.

Since our results have considerable implications both scientifically and practically, we conducted a second study to once again test our hypotheses and check whether the results of Study 1 are robust with a bigger sample size and, thus, more statistical power. In addition, we include potential moderator variables to test whether leader-related or organizational variables that may facilitate or even reward exemplary followership can foster the relationship of followership with job attitudes.

## STUDY 2

Compared to Study 1, we made the following changes in Study 2: In addition to the variables that were considered in Study 1, we also tested two moderator variables that were not part of Study 2 (*i.e.*, leader humility and perceived organizational support, see below). Thus, Study 2 went beyond a mere replication of Study 1 and examined two potential new conditions under which the relationships of followership with job attitudes may be fostered. Finally, we used a shorter time-lag than in Study 1 and assessed whether the followers' direct leader (*i.e.*, the person they referred to when responding to the followership questionnaire) had changed between t1 and t2 to rule out potential shortcomings of our initial study design.

### Followership, leader humility, and perceived organizational support

We argue that leader humility is likely to moderate the proposed relationships between followership and job attitudes. Leader humility comprises leaders' willingness to view oneself accurately, teachability, and the appreciation of the followers' strengths and contributions (*Chiu, Owens & Tesluk, 2016*; *Owens, Johnson & Mitchell, 2013*). Specifically, *Kelley (1992)* suggests that both actively engaged and independent, critical followers are most valuable for leaders and for organizations. However, he admits that not every leader or organization might actually appreciate active and/or critical followership (*Kelley, 1988*). Thus, positive relationships of followership behavior with job satisfaction and organizational commitment is likely to depend on leaders' teachability and leaders' appreciation for ''exemplary followership'' as a valuable resource. That is, followership can only unfold its positive effect on followers' job attitudes when leaders' are willing to learn from followers and appreciate their contributions. Therefore, we expect that high leader's humility facilitates the relationship between followership (at t1) and job attitudes (at t2). Thus, we predict:

*Hypothesis 12: AE at t1 will be more positively related to (a) job satisfaction at t2 and (b) organizational commitment at t2 when the leader's humility at t1 is high, as compared to when leader humility at t1 is low.*

*Hypothesis 13: ICT at t1 will be less negatively related to (a) job satisfaction at t2 and (b) organizational commitment at t2 when the leader's humility at t1 is high, as compared to when leader humility at t1 is low.*

In addition to leader humility, we expect that perceived organizational support (POS) moderates the relationship of followership behaviors and job attitudes as another form of appreciation for ''exemplary followership'' as a valuable resource. This is generally

consistent with notions from Organizational Support Theory according to which "employees develop global beliefs concerning the extent to which the organization values their contributions and cares about their well-being" (*Rhoades & Eisenberger, 2002*, p. 698). Organizational Support Theory (*Eisenberger et al., 1986*) assumes that the caring, approval, and respect connoted by POS strengthen the employees' beliefs that the organization recognizes and rewards increased performance (*Rhoades & Eisenberger, 2002*). Meta-analytic results by *Kurtessis et al. (2017)* suggest that POS is an important link between favorable treatment by the organization and employees' positive orientation toward the organization, psychological well-being, and performance. For example, they report positive relationships of POS with employees' performance–reward expectancy, commitment, and job satisfaction (*Kurtessis et al., 2017*). In line with the general predictions of Organizational Support Theory and these meta-analytical results, the interaction of the efforts of carrying out an active and independent follower role and its acknowledgement through POS (at t1) should also be positively related to job satisfaction and organizational commitment (at t2). Thus, we predict:

*Hypothesis 14: AE at t1 will be more positively related to (a) job satisfaction at t2 and (b) organizational commitment at t2 when POS at t1 is high, as compared to when POS at t1 is low.*

*Hypothesis 15: ICT at t1 will be less negatively related to (a) job satisfaction at t2 and (b) organizational commitment at t2 when POS at t1 is high, as compared to when POS at t1 is low.*

In addition, we will also investigate as a research question whether the simultaneous occurrence of leader humility and POS (at t1) will strengthen the relationships between followership behaviors (at t1) and job attitudes (at t2), since the acknowledgement of "exemplary followership" might be even more present for the follower, when both the leader and the organization accordingly consider active and critical followership a valuable resource.

## MATERIALS AND METHODS

We again conducted a two-wave survey. In contrast to the time lag between the two measurement waves in Study 1 (*i.e.,* nine to 12 months), we realized a time lag of four months for Study 2. With a shorter time lag, we intended to rule out possible shortcomings of Study 1: Some effects might have simply disappeared over time and thus could not be detected. This would be the case, for instance, if the respondents' leader changed between the two surveys, if the proposed effects unfolded too quickly, or if the effects did not last long enough to be detected (for a related analysis of time-lags in leadership research, see *Fischer, Dietz & Antonakis, 2017*). Since there were no other longitudinal studies that studied followership behaviors when we planned our study, we drew from the meta-analysis on relationships between performance and job attitudes by *Riketta (2008)* to obtain information about suitable time lags. On that basis, we considered four months to be a reasonable time lag to detect effects on attitudinal variables in the context of work. This was also in line with the general recommendation of pertinent

[6]Our research is in line with the ethical principles of the Declaration of Helsinki and the Psychologists Code of Conduct of the American Psychological Association. We obtained ethical approval of the University of Münster for Study 2 (approval number 2022-65-ChN).

literature to use, in case of doubt, a rather shorter time-lag (*e.g.*, *Dwyer, 1983*; *Griep et al., 2021*; *Voelkle et al., 2012*). This study was preregistered at the Open Science Framework (https://osf.io/9q48p/?view_only=b02c98f57f7841129cf6daf42405b410).

## Sample

We conducted an anonymous online survey *via* the German "WiSoPanel", an online-access-panel based on voluntary registration (*Göritz, 2014*). All eligible users were invited to participate in 2021 and 2022. We included only employees with a direct superior to ensure credible responses concerning followership behaviors in organizations. Participants gave their consent for participation within the questionnaire by checking the associated box.[6] At t1, 777 respondents completed the questionnaire. However, two cases were excluded because of the respondents' statement not to use the data. Thus, 775 respondents of t1 were invited to participate again at t2. At t2, 628 respondents completed the questionnaire (response rate 81%), while three cases were excluded due to the respondents' statement not to use the data. Following our preregistration, we additionally tried to detect careless responses by following the procedures recommended by *Meade & Craig (2012)*. We tested for zero-within-variance in responses and defined the values in question as missing values. For t1 data, 70 cases were affected regarding leader humility, 28 cases were affected regarding followership, 46 cases were affected regarding job satisfaction and self-efficacy, and 41 cases were affected regarding POS and organizational commitment. For t2 data, 69 cases were affected regarding leader humility, 25 cases were affected regarding followership, 33 cases were affected regarding job satisfaction and self-efficacy, and 39 cases were affected regarding POS and organizational commitment. Furthermore, we computed Mahalanobis Distance overall items and excluded 55 cases that were detected as outliers.

The final sample consisted of 570 employees. The mean age was 49.1 years and 55% of the respondents were female, 45% were male. The level of education was distributed as follows: 1% had no professional qualification, 42% had a completed apprenticeship, 22% had a university of applied science degree, and 35% had a university degree, and 1% had another degree that was not specified in the questionnaire. Nearly a third (34%) of the respondents worked in the public service sector, 5% worked in the finance service sector (banking or insurance), and 61% worked in another sector (*i.e.*, for instance, other professional services or industry sector). Hence, in Study 2, we tested our hypotheses in a more heterogeneous sample than in Study 1.[7]

## Measures

We used the exact same measures that were used in Study 1 for followership, job attitudes and self-efficacy. Additionally, we measured leader humility with the "Expressed Humility Scale" of *Owens, Johnson & Mitchell (2013)* as adapted for the leadership context by *Chiu, Owens & Tesluk (2016)*. A sample item was "My supervisor shows appreciation for the unique contributions of others" (nine items in total). Response options ranged from *strongly disagree (1)* to *fully agree (7)*. Following *Bracken & Barona (1991)*, this scale was translated into German and back-translated into the original language (English) by another organizational psychology expert without prior knowledge of the questionnaire. Finally,

[7]Note that we preregistered our data collection with the intention to generate a sample that was comparable to the sample of Study 1. Since the panel was not successful in providing the desired highly comparable distribution for the sectors, we had to use the resulting, more heterogeneous sample for Study 2.

this back-translation was compared to the original version by a native speaker. POS was be measured by *Eisenberger et al. (2001)* in the German translation by *Klasmeier & Rowold (2020)*. A sample item was "The organization values my contributions to its well-being" (six items in total). Response options ranged from *strongly disagree (1)* to *fully agree (7)*. All questions were in German language.

## Data analysis strategy

We applied the same statistical analyses that we used in our initial study. In order to investigate the construct stability of followership behavior, a latent state-trait analysis (*Geiser, 2020*; *Steyer, Schmitt & Eid, 1999*) was conducted. Furthermore, we again applied LACS models (see *Geiser, 2020*) to test the hypotheses on the interrelations of followership with the other study variables. For the models that tested the interaction effects of AE and ICT, we again controlled for the autoregressive effects of job attitudes and self-efficacy.

## Pre-analyses

*Construct validity.* In order to ensure the distinctiveness of our study variables, we applied Chi square difference tests to test whether our measurement model fitted the data better than two alternative models both at t1 and t2. In the first alternative model, we specified both job attitudes (*i.e.,* job satisfaction and organizational commitment) as one common factor. Our model fitted the data better than the alternative model both at t1, $\chi^2_{\text{diff}}(5) = 347.902$, $p < .001$, and t2, $\chi^2_{\text{diff}}(5) = 290.357$, $p < .001$. In the other alternative model, we specified both organization-related variables (*i.e.,* organizational commitment and POS) as one common factor. Again our model fitted the data better at t1, $\chi^2_{\text{diff}}(5) = 186.424$, $p < .001$, and t2, $\chi^2_{\text{diff}}(5) = 165.31$, $p < .001$. Hence, our analyses indicate the distinctiveness of our study variables.

*Measurement equivalence across time.* Before testing the hypotheses, we again tested for measurement equivalence across time. According to the tests of measurement equivalence, the strict model was preferred for AE, ICT, job satisfaction, organizational commitment. The strong model was preferred for self-efficacy. Hence, all constructs that were used for longitudinal analysis showed at least strong measurement equivalence (see SM1 for more details).

## RESULTS

Table 2 shows the descriptive statistics, the intercorrelations of the latent variables and the reliabilities of the measures of Study 2.

## Temporal consistency of followership behavior

The results of Study 2 again revealed that AE was rather stable and trait-like over time as indicated by consistencies exceeding the commonly applied 50% threshold for both AE at time 1 (Con[ $\tau_{\text{AE},t1}$] $= .73$, $p < .001$,) and AE at time 2 (Con[ $\tau_{\text{AE},t2}$] $= .80$, $p < .001$) as compared to the coefficients that indicate occasion specificity (Osp[ $\tau_{\text{AE},t1}$] $= .27, p < .001$; Osp[ $\tau_{\text{AE},t2}$] $= .20$, $p < .001$). The model fitted the data well, $\chi^2(\text{df}) = 277.90(151)$,

**Table 2  Descriptives, intercorrelations between latent variables, and internal consistencies of Study 2.**

| Model | M | SD | 1 | 2 | 3 | 4 | 5 | 6 | 7 | 8 | 9 | 10 | 11 | 12 | 13 | 14 |
|---|---|---|---|---|---|---|---|---|---|---|---|---|---|---|---|---|
| 1. Followership: AE (t1) | 5.10 | 1.07 | .99″ | | | | | | | | | | | | | |
| 2. Followership: ICT (t1) | 4.61 | 1.08 | .54*** | .84″ | | | | | | | | | | | | |
| 3. Job satisfaction (t1) | 5.02 | 1.21 | .54*** | .22*** | .91′ | | | | | | | | | | | |
| 4. Organizational commitment (t1) | 4.79 | 1.45 | .46*** | .08 | .73*** | .80′ | | | | | | | | | | |
| 5. Self-efficacy (t1) | 5.31 | 1.11 | .71*** | .57*** | .61*** | .38*** | .93′ | | | | | | | | | |
| 6. Perceived organizational support (t1) | 4.34 | 1.37 | .45*** | .16** | .70*** | .84*** | .44*** | .95′ | | | | | | | | |
| 7. Leader humility (t1) | 4.70 | 1.42 | .49*** | .19*** | .66*** | .59*** | .38*** | .60*** | .91′ | | | | | | | |
| 8. Followership: AE (t2) | 5.08 | 1.01 | .80*** | .30*** | .41*** | .41*** | .58*** | .40*** | .38*** | .99″ | | | | | | |
| 9. Followership: ICT (t2) | 4.68 | 1.05 | .50*** | .69*** | .17** | .13* | .44*** | .18*** | .12* | .61*** | .85″ | | | | | |
| 10. Job satisfaction (t2) | 5.09 | 1.16 | .47*** | .15*** | .77*** | .64*** | .53*** | .58*** | .55*** | .54*** | .31*** | .90′ | | | | |
| 11. Organizational commitment (t2) | 4.73 | 1.42 | .42*** | .04 | .63*** | .92*** | .34*** | .72*** | .56*** | .49*** | .23*** | .77*** | .79′ | | | |
| 12. Self-efficacy (t2) | 5.34 | 1.12 | .56*** | .35*** | .45*** | .32*** | .81*** | .33*** | .27*** | .69*** | .54*** | .60*** | .40*** | .93′ | | |
| 13. Perceived organizational support (t2) | 4.31 | 1.34 | .37*** | .15*** | .58*** | .72*** | .35*** | .79*** | .56*** | .45*** | .26*** | .72*** | .85*** | .38*** | .95′ | |
| 14. Leader humility (t2) | 4.65 | 1.41 | .43*** | .09 | .53*** | .54*** | .32*** | .51*** | .75*** | .48*** | .23*** | .68*** | .63*** | .31*** | .63*** | .92′ |

**Notes.**

$N = 570$.

AE, Active engagement; ICT, Independent, critical thinking.

Values along the diagonal represent internal consistency (′$\omega$ or ″$\omega$s).

*$p < .05$.
**$p < .01$.
***$p < .001$.

Ribbat et al. (2023), *PeerJ*, DOI 10.7717/peerj.16135

$p < .001$, $\chi 2/ df = 1.84$, RMSEA $= .04$, CFI $= .98$, SRMR $= .04$. Hence, our results suggest that AE was consistent across time and rather trait-like, which supported Hypothesis 1a.

The results of the model for ICT revealed Con[ $\tau_{\text{ICT},t1}$ ] $= .58$, $p < .001$, Con[ $\tau_{\text{ICT},t2}$ ] $= .63$, $p < .001$, Osp[ $\tau_{\text{ICT},t1}$ ] $= .42$, $p < .001$, and Osp[ $\tau_{\text{ICT},t2}$ ] $= .37$, $p < .001$. The model fit was good: $\chi^2(\text{df}) = 36.25(24)$, $p = .05$, $\chi 2/ df = 1.51$, RMSEA $= .03$, CFI $= .99$, SRMR $= .04$. Therefore, ICT was consistent across time and rather trait like. Hence, Hypothesis 1b was also supported.

## Followership and job attitudes

In the first cross-lagged model, the relationship of AE with job satisfaction was explored. Both autoregressive paths were significant (for AE: $\beta = .80$, $p < .001$; for job satisfaction: $\beta = .67$, $p < .001$). The relationship of job satisfaction (at t1) with AE (at t2) was not significant, $\beta = -.03$, $p = .26$, thereby not supporting Hypothesis 4a in Study 2. However, AE (at t1) was positively related to job satisfaction (at t2), $\beta = .09$, $p = .02$, thereby supporting Hypothesis 2 in Study 2. The model explained 59% of the variance in AE and 52% of the variance in job satisfaction and the model fit was good: $\chi^2(\text{df}) = 634.05(381)$, $p < .001$, $\chi 2/ df = 1.66$, RMSEA $= .03$, CFI $= .98$, SRMR $= .04$. We received similar results when controlled for a potential change of the reference leader since t1 in addition to the autoregressive paths. Results were then $\beta = .08$, $p = .04$, for the relationship between AE (at t1) and job satisfaction (at t2), and $\beta = -.03$, $p = .26$, for the opposite path.

The relationship of AE and organizational commitment was explored in the second cross-lagged model. Both autoregressive paths were significant (for AE: $\beta = .75$, $p < .001$; for organizational commitment: $\beta = .86$, $p < .001$). The relationship of AE (at t1) with organizational commitment (at t2) was not significant, $\beta = -.05$, $p = .12$, thereby not supporting Hypothesis 2b. Organizational commitment (at t1) was positively related to AE (at t2), $\beta = .08$, $p = .02$, thereby supporting Hypothesis 4b. The model explained 60% of the variance in AE and 72% of the variance in organizational commitment and the model fit was good: $\chi^2(\text{df}) = 432.59(249)$, $p < .001$, $\chi 2/ df = 1.74$, RMSEA $= .04$, CFI $= .98$, SRMR $= .04$. We again received similar results when we additionally controlled for a potential leader change (for the relationship between AE (at t1) and organizational commitment (at t2): $\beta = -.04$, $p = .13$; for the opposite path: $\beta = .08$, $p = .02$).

The third cross-lagged model explored the relationship of ICT with job satisfaction. Both autoregressive paths were significant (for ICT: $\beta = .61$, $p < .001$; for job satisfaction: $\beta = .72$, $p < .001$). We could neither find a significant relation of job satisfaction (at t1) to ICT (at t2), $\beta = .03$, $p = .23$, nor between ICT (at t1) and job satisfaction (at t2), $\beta = .01$, $p = .43$. Hence, these results do not support Hypotheses 3a and 5a. The model explained 38% of the variance in ICT and 52% of the variance in job satisfaction and the model fit was again good: $\chi^2(\text{df}) = 196.28(146)$, $p = .004$, $\chi 2/ df = 1.34$, RMSEA $= .03$, CFI $= .99$, SRMR $= .03$.

The fourth cross-lagged model included ICT and organizational commitment. Both autoregressive paths were significant (for ICT: $\beta = .62$, $p < .001$; for organizational commitment: $\beta = .84$, $p < .001$). No significant relationship of ICT (at t1) with organizational commitment (at t2) could be found, $\beta = -.03$, $p = .18$. Organizational

[8]Note that we did not preregister Hypotheses 6 and 7 for Study 2, because our results of Study 1 pointed in the opposite direction than we had expected. However, we obtained different results in Study 2. Hence, we tested all of our hypotheses again in Study 2.

commitment (at t1) was also not related to ICT (at t2), $\beta = .02$, $p = .34$. These results do not support Hypotheses 3b and 5b. The model explained 39% of the variance in ICT and 72% of the variance in organizational commitment and the model fit was good: $\chi^2(\text{df}) = 134.32(68)$, $p < .001$, $\chi^2/df = 1.98$, RMSEA $= .04$, CFI $= .98$, SRMR $= .04$.

In Hypotheses 6 and 7, we proposed the relationships of AE (at t1) and ICT (at t1) with job attitudes (at 2) to be stronger than those of job attitudes (at t1) with AE (at t2) and ICT (at t2).[8] While we found a significant relationship for AE (at t1) and job satisfaction (at t2) but not for the opposite path, the results of Study 2 support Hypothesis 6a. However, we could not find significant relationships between AE (at t1) and organizational commitment (at t2) or of ICT (at t1) with any job attitude. Therefore, Hypothesis 6b and Hypothesis 7 were not supported. Furthermore, in Hypothesis 8 we posited ICT (at t1) to be less negatively related to job satisfaction and organizational commitment (at t2) when AE (at t1) was high, as compared to when AE (at t1) was low. In Study 2, the interaction of AE (at t1) and ICT (at t1) was not a significant predictor for organizational commitment (at t2), $\beta = .01$, $p = .37$. In addition, the interaction (at t1) did not predict job satisfaction (at t2), $\beta = .01$, $p = .36$. Thus, Hypothesis 8 was not supported.

### Followership and self-efficacy

We also explored the relationship of AE and ICT with self-efficacy in different cross-lagged models. The first cross-lagged model included AE and self-efficacy. We could not find a significant relationship of AE (at t1) with self-efficacy (at t2), $\beta = -.07$, $p = .09$, nor did self-efficacy (at t2) predict AE (at t1), $\beta = -.03$, $p = .55$. Note, however, that we applied a two-tailed test for significance in this model, since the regression coefficient pointed in a different direction than we had expected. Hence, Hypotheses 9a and 11a were not supported. The autoregressive paths were significant (for AE: $\beta = .78$, $p < .001$; for self-efficacy: $\beta = .83$, $p < .001$). The model explained 61% of the variance in AE and 64% of the variance in self-efficacy with a good model fit: $\chi^2(\text{df}) = 628.73(375)$, $p < .001$, $\chi^2/df = 1.68$, RMSEA $= .04$, CFI $= .98$, SRMR $= .04$.

The relationship of ICT and self-efficacy was explored in another cross-lagged model. The autoregressive paths were again significant (for ICT: $\beta = .59$, $p < .001$; for self-efficacy: $\beta = .85$, $p < .001$). We could not find a significant relationship between self-efficacy (at t1) with ICT (at t2), $\beta = .05$, $p = .34$, which does not support Hypothesis 11b. However, ICT (at t1) was negatively related to self-efficacy (at t2), $\beta = -.12$, $p = .004$. We again applied a two-tailed test for this model, since we had predicted this relationship to be positive. Hence, Hypothesis 9a was also not supported. The model explained 39% of the variance in ICT and 63% of the variance in self-efficacy with a good model: $\chi^2(\text{df}) = 211.12(140)$, $p < .001$, $\chi^2/df = 1.51$, RMSEA $= .03$, CFI $= .99$, SRMR $= .03$. We received similar results when we additionally controlled for a potential leader change (for the relationship between ICT (at t1) and self-efficacy (at t2): $\beta = -.12$, $p = .004$; for the opposite path, $\beta = .05$, $p = .34$).

In addition, we tested whether the interaction of AE (at t1) and ICT (at t1) could predict self-efficacy (at t2). The interaction of AE (at t1) and ICT (at t1) did not predict self-efficacy (at t2), $\beta = .05$, $p = .12$. Thus, Hypothesis 10 was not supported.

## Followership, leader humility and POS

In Study 2, we tested whether leader humility or POS could be mechanisms that foster the relationships of AE or ICT with job attitudes. The interaction of AE and leader humility (at t1) did not predict job satisfaction (at t2), $\beta = .00$, $p = .46$, and it did not predict organizational commitment (at t2), $\beta = -.02$, $p = .31$. Hence Hypothesis 12 was not supported. The interaction of ICT and leader humility (at t1) did neither predict job satisfaction (at t2), $\beta = .01$, $p = .43$, nor organizational commitment (at t2), $\beta = .03$, $p = .22$. Thus, Hypothesis 13 was not supported. Furthermore, we could neither find a significant relationship of the interaction of AE and POS (at t1) with job satisfaction (at t2), $\beta = .05$, $p = .08$, nor with organizational commitment (at t2), $\beta = .03$, $p = .17$. The interaction of ICT and POS (at t1) could neither predict job satisfaction (at t2), $\beta = -.03$, $p = .20$, nor organizational commitment (at t2), $\beta = -.02$, $p = .24$. Therefore, Hypotheses 14 and 15 were not supported. Finally, a three-way interaction of AE, leader humility and POS (at t1) could not predict job satisfaction (at t2), $\beta = -.01$, $p = .43$, nor organizational commitment (at t2), $\beta = .03$, $p = .08$. A three-way interaction of ICT, leader humility and POS did neither predict job satisfaction (at t2), $\beta = .04$, $p = .07$, nor organizational commitment (at t2), $\beta = .01$, $p = .35$.[9]

[9] Note that Mplus did not provide standardized coefficients for the three-way interaction models. Thus, for these models, we report unstandardized coefficients.

# DISCUSSION

The results of both of our studies support *Kelley*'s (*1992*) assumption that followership behaviors are rather stable behavior patterns. We found significant relations of job attitudes with active, engaged followership behavior in the cross-lagged models, above and beyond the autoregressive effects in both studies. We also found significant relations of AE (at t1) with job satisfaction (at t2) and of ICT (at t1) with self-efficacy (at t2) in Study 2. The interaction of AE and ICT (at t1), however, predicted organizational commitment (at t2) only in Study 1.

Although the regression coefficients ($.08 \leq \beta \geq .14$) can be described as rather small (*Cohen, 1988*), the identified effects indicate important findings for three reasons. First, cross-lagged effects are generally hard to find. That is, autoregressive effects are often strong and therefore explain much of the variance by themselves (*Adachi & Willoughby, 2015*; *Geiser, 2020*). Hence, there is often little variance left to be explained by the cross-lagged effects. Second, since our results demonstrate that followership styles are rather stable behavior patterns, predicting change in followership behavior is even more noteworthy. Third, the reported effect sizes are indeed in a typical range as compared to similar studies with cross-lagged models (see, for instance, *Riketta, 2008*; *Sonnentag, Binnewies & Mojza, 2010*). However, across both studies, we found mixed results regarding our hypotheses. We will elaborate on these findings below.

# THEORETICAL IMPLICATIONS

Our studies contribute to a better understanding of followership (*Kelley, 1992*) and its outcomes (*Uhl-Bien et al., 2014*). To go beyond prior research, we tested *Kelley*'s (*1992*) followership behaviors longitudinally, and explored their relationships to critical

job attitudes (*i.e.,* job satisfaction and organizational commitment) and self-efficacy as potential predictors and consequences. Our findings support *Kelley*'s (*1992*) assumption that followership styles (*i.e.,* AE and ICT) are rather stable behavior patterns. This is an important finding, because a more state-like nature of followership behavior would necessarily shift the focus of future research from general and stable factors to more specific, situational, and contingent factors as followership would then rather be spontaneous, dynamic, or variable. For instance, the common perspective on leadership styles (see *Anderson & Sun, 2017*; *Bass & Avolio, 1995*) as consistent behavior patterns of leaders has been challenged most recently. An increasing number of authors argue for considering leader behaviors as more dynamic (see, for instance, *Kelemen, Matthews & Breevaart, 2020*; *McClean et al., 2019*). In contrast, our findings suggest that an opposite perspective is adequate for *Kelley*'s (*1992*) followership styles. However, several studies could show how leader and follower identities (that did not necessarily correspond to their formal ranks as "leaders" and "subordinates") were (co-)constructed in different leadership situations (*e.g.,* *Blom & Alvesson, 2014*; *Larsson & Nielsen, 2021*; *Van De Mieroop, 2020*). Adopting the co-construction approach to followership (see *Uhl-Bien et al., 2014*), *Larsson & Nielsen (2021)*, for instance, found that leader and follower roles remained abstract in the workplace interactions that they analyzed. Participants rather focused on negotiated, task-oriented, expert or non-expert identities (*Larsson & Nielsen, 2021*). Hence, a more dynamic or situational perspective on followership can be helpful in those contexts, in which leaders and followers are not determined by their position. Our findings, however, suggest that subordinates tend to enact their follower role rather consistently within a continuum between active and passive, independent and uncritical (see *Kelley, 1992*), when they interact with their superior leader.

It is noteworthy and plausible, however, that ICT is less stable or trait-like than AE, a finding that we consistently observed across our studies. Speaking up in front of the leader can be a risky behavior for followers, particularly when it is done in challenging rather than supportive ways (*Burris, 2012*). Thus, followers probably consider their concerns carefully before they actually express them to the leader (*Bashur & Oc, 2015*; *Detert et al., 2013*). Consequently, they might hold back their contrary view, when they feel that their concern is less important, but they rather have the courage to speak up in situations where they perceive urgency. Therefore, it is plausible that potentially risky critical followership is more dependent on the evaluation of the urgency and appropriateness of a specific situation than the willingness to support the leader through active engagement. Still, even if followership behaviors cannot be seen as fully invariable personality traits, both AE and ICT are more trait-like than state-like according to our findings.

With regard to the relationships of AE and ICT with critical job attitudes (*i.e.,* job satisfaction and organizational commitment) and self-efficacy, we obtained mixed results. In Study 1, we did not find significant relationships of AE or ICT (at t1) with job attitudes (at t2) that we had predicted. Furthermore, we did not detect any significant relation of followership with self-efficacy. In contrast, job satisfaction and organizational commitment (at t1) were positively related to AE (at t2). In Study 2, we obtained similar results with three exceptions. First, in Study 2, we did not find a relationship of job satisfaction (at

t1) with AE (at t2), but AE (at t1) predicted later job satisfaction (at t2). Second, ICT (at t1) was negatively related to self-efficacy (at t2) in Study 2, although we had predicted a positive relation. This relationship was not significant in Study 1. Third, our results of Study 1 suggest that high levels of both AE and ICT (*i.e.,* an exemplary followership style, see *Kelley, 1992*) lead to higher organizational commitment, thereby supporting *Kelley*'s (*1992*) assumption that high levels of both AE and ICT imply desirable organizational behaviors. However, we could not detect this interaction effect in Study 2. One reason for the relatively low bivariate associations across time and for the missing interaction effects in our studies could be that our analysis was restricted by low variance in followership styles. Most participants in both studies adopted either the pragmatist or the exemplary followership style (see SM2 for more details). Furthermore, the high mean values of both followership dimensions in both studies (see Tables 1 and 2) might indicate a certain risk of social desirability of the questions, which has also been discussed in previous studies (see, for instance, *Gatti et al., 2014*; *Ribbat, Krumm & Hüffmeier, 2021*).

While some of our results indicate that followership behaviors can predict later job attitudes or self-efficacy, most of our results either point in the opposite direction or indicate no significant relationships at all. Those results challenge the original idea of followership theory (*Kelley, 1992*; *Uhl-Bien et al., 2014*) that followership behaviors are significant predictors for organizational variables. However, our studies do provide at least a few hints for the potential of followership behaviors as predictors for job attitudes and self-efficacy. In sum, our findings raise important questions for future followership research.

First, the mixed findings across our two studies suggest that future research needs to elaborate on time lags in longitudinal research. An increasing number of authors calls for the appropriate inclusion of time aspects both in research designs and in theory, which has been neglected in leadership and organizational research for a long time (see *Fischer, Dietz & Antonakis, 2017*; *Griep & Zacher, 2021*; *Griep et al., 2021*; *Shamir, 2011*). In Study 1, we used data from a preexisting, more comprehensive data set that had a time lag of nine to 12 months. Since there are some comparable studies that used shorter time lags to detect interrelations of job attitudes and behaviors (for an overview, see, for instance, *Riketta, 2008*), we used a shorter time lag (*i.e.,* four months) in Study 2. We indeed found two relationships of followership behaviors (at t1) with later job satisfaction and self-efficacy (at t2) that we did not detect in Study 1. These findings suggest that followership behaviors might affect job attitudes or self-efficacy in the short-term rather than in the long-term. Hence, our findings correspond to the analysis of leadership research by *Fischer, Dietz & Antonakis (2017)* in view of two points: *Fischer, Dietz & Antonakis (2017)* concluded that effects at the team- or individual level unfold rather quickly, while they do not last very long. Furthermore, they stated that effects on behaviors typically take longer to unfold and persist longer than do effects on cognitions or emotions (*Fischer, Dietz & Antonakis, 2017*). Still, we need more information on the role that time plays both in followership and leadership research to develop theory further and to better understand the nature of the studied effects (*Castillo & Trinh, 2018*; *Griep et al., 2021*). We, therefore, highly recommend future research to consider different time lags in order to learn in which time

frames the effects of followership behaviors occur and when they potentially decline. In our two studies, we could demonstrate that followership behaviors are relatively stable, thereby supporting *Kelley*'s (*1992*) conceptions of rather consistent followership styles. Hence, our findings can guide future followership research, since "the temporal stability of variables and the stability of effects on these variables are criteria for deciding on repeated-measures designs" (*Fischer, Dietz & Antonakis, 2017*, p. 1740).

Second, we encourage future research to explore potential mechanisms that might appear to be the missing links for the relationships we could not detect in our studies. *Benson, Hardy & Eys (2015)*, for instance, suggest that the situational context affects how leaders see followership behaviors. That is, attempts to influence a leader's decisions in front of others might not be appreciated by the leader. However, while this might be obvious for independent and critical behavior, we would still expect the leader to acknowledge the follower's support through active engagement. *Hoption (2016)*, for instance, demonstrated that the followers' provision of help to leaders corresponds to greater leader relationship satisfaction. Thus, it is somewhat surprising that we could find a positive relation of AE (at t1) with job satisfaction (at t2) only in Study 2, and no positive relation of AE (at t1) with organizational commitment (at t2) in both of our studies. Future research should, therefore, explore potential moderator and mediator variables that might uncover the most detrimental and beneficial conditions for positive followership outcomes.

We suggest that the leaders' preferences and reactions to certain followership styles should be a good starting point to detect the missing links between active and critical followership and followership outcomes. We firstly tested leader humility, which was not confirmed as a moderator in our study despite our prediction. However, *Shen & Abe (2022)*, for instance, found an indirect effect of followership behaviors on job performance through perceived supervisor support. Moreover, we expected POS to strengthen the employees' beliefs that the organization recognizes and rewards increased performance (*Rhoades & Eisenberger, 2002*) and thereby foster the relationships between active and critical followership with job attitudes. The corresponding hypotheses, however, were not supported by our data. We could neither find a three-way interaction of followership behavior, leader humility and POS. One reason for this could be that leader humility, and POS were highly correlated ($r = .60$; see Table 2). Future research could further explore which organizational environments affect followership behaviors and its potential outcomes. *Blair & Bligh (2018)*, for instance, argue that values and norms focused on hierarchy and control limit active follower beliefs in shared responsibility for leadership. The findings of *Carsten et al. (2010)* echo this argument by suggesting that followers' ability to take initiative is diminished by strong bureaucracy.

Third, we explored the relationship of followership with self-efficacy for the first time. In our hypotheses, we postulated that the followers' self-efficacy substantially results from ascribing (at least parts of) the leadership success to own efforts when carrying out an active and independent follower role as they experience mastery in this way. According to Social Cognitive Theory (*Bandura, 1997*), mastery experience is considered to have the strongest impact on the development of self-efficacy. We could only detect a negative relation of ICT (at t1) with later self-efficacy (at t2) in Study 2, although we had predicted

a positive relation. One reason could be again that ICT was not appreciated by the leaders. The followers' efforts to contribute independently then might had been not successful and, thus, they were experienced as a personal failure rather than personal mastery. In addition, we did not detect any significant relationship of self-efficacy (at t1) with later followership behaviors (at t2), which might result from strong autoregressive paths or inconvenient time-lags. Future research should further investigate whether followership can predict self-efficacy or vice versa with different time-lags. Thereby, research should also clarify whether or under which circumstances exemplary followership implies mastery experience and to which extent Social Cognitive Theory (*Bandura, 1997*) is applicable to followership research. Moreover, future studies could additionally build on other sources for self-efficacy. If other sources than mastery experience played a more important role than we had expected based on Social Cognitive Theory (*Bandura, 1997*), different effects could have cancelled each other out. This would be the case, for instance, if simply taking directions (*i.e.,* uncritical followership) led to vicarious experience for some followers, while high levels of independent thinking led to mastery experience for others.

## PRACTICAL IMPLICATIONS

Our findings provide interesting information for followers, leaders, and organizations. We could demonstrate that followership behaviors are relatively stable behavior patterns, even though it is possible that they change under certain circumstances. Thus, *Kelley*'s (*1992*) "Identify Your Followership Style Questionnaire" indeed allows for assessing one's own general behavioral tendencies and preferences regarding the interaction with the leader. Hence, our findings support *Kelley*'s (*1992*) conceptual idea of followership styles in this respect. Since followership, here, is enacted from a formal follower rank or position (see also *Uhl-Bien et al., 2014*), followership styles should be particularly relevant in the contexts of hierarchical organizations. It may be more difficult to "Identify Your Followership Style", when the reference leader is not apparent in non-hierarchical organizational structures. This would be the case, for instance, in self-organized or agile teams, in which leader and follower roles shift more fluently (see, for instance, *Srivastava & Jain, 2017*; *Zhu et al., 2018*).

Our finding regarding the consistency of *Kelley*'s (*1992*) followership behaviors is valuable for followers, because knowing one's own style allows for reflecting on how the style might fit or not with the leader and/or organization. This is important, because many studies have demonstrated that person-organization-fit, for example, increases satisfaction and reduces the employee's intention to leave the organization (*Verquer, Beehr & Wagner, 2003*; *Jin, McDonald & Park, 2018*). Furthermore, person–supervisor fit is associated with greater leader satisfaction and with a better relationship between leader and follower (*Kristof-Brown, Zimmerman & Johnson, 2005*; *Marstand, Martin & Epitropaki, 2016*). For leaders, the assessment of the followers' individual styles may help to understand why followers tend to behave in certain ways. This, in turn, can facilitate an adequate handling of leader-follower-interactions.

For organizations, recognizing followership styles as rather stable behavioral patterns can help to inspire development programs for followership styles that are needed or preferred

within the organization. An increasing number of authors calls for such development programs (see, for instance, *Bufalino, 2018*; *Hoption, 2014*; *Logan & Ganster, 2007*). While complementing effective leadership training with followership development programs (*Bufalino, 2018*; *Hoption, 2014*) might be a useful approach to foster desired followership behaviors, our findings suggest that it might not be sufficient to simply call for active and/or critical followership to obtain positive organizational outcomes. According to our study, satisfied and committed followers are more likely to participate actively in the leadership process. Thus, our findings at least indirectly point to satisfactory working conditions and appropriate organizational goals as beneficial settings for active followership.

Following *Andersson*'s (*2018*) argumentation that followership is an important social resource for organizational resilience, organizations might care about developing desired followership to successfully meet future challenges that could emerge from a pandemic or from other far-reaching developments like, for instance, technological change, disruptive innovation, or climate change. Consequently, followers, leaders, and organizations should reflect on what styles of followership they want to have or show, and which styles they need. *Kelley*'s (*1992*) operationalization of followership behaviors (*i.e.,* AE and ICT) can be a useful tool to learn about the way followers actually carry out their follower role. Still, there is more research needed to better understand the impact that different followership styles can have both on the individual and on the organizational level.

## LIMITATIONS

There are several limitations to this study. First, we used only self-report data, which implies certain risks of common-method bias (*Podsakoff et al., 2003*). We addressed related problems by using a time-lagged design. However, future research could complement the evaluations of the followers with their leader's perceptions or with independent observations to further decrease such risks (for related recommendations, see also *Gatti et al., 2014*; *Ribbat, Krumm & Hüffmeier, 2021*).

Second, an increasing number of authors (see, for instance, *Antonakis et al., 2010*; *Bastardoz et al., 2023*; *Wulff et al., 2023*) pointed to the threat of making false causal claims due to endogeneity problems. Since our analysis relied on self-assessment assessed with survey measures both for behaviors and attitudes, our analyses were at risk to suffer from such endogeneity problems. However, the inclusion of multiple measurement points and the cross-lagged design should have reduced the risks for simultaneity bias and reverse causality in our studies. Still, causal inferences should be made with due caution. Future studies, therefore, could apply a variety of methods to further test the causal relationships between followership behaviors and followership outcomes (*e.g.,* experimental designs or multi-method designs with independent observations).

Third, we used data from different samples that had been collected within different time-lags. We used data from a preexisting more comprehensive data set that had a time lag of nine to 12 months in our first study. Since we assumed that a positive impact of followership may have vanished by this time and since there are some comparable studies that used shorter time lags to detect interrelations of job attitudes and behaviors (for an

overview, see, for instance, *Riketta, 2008*), we used a shorter time-lag (*i.e.,* four months) in our second study. While our findings suggest that followership behaviors affect job attitudes and self-efficacy rather short-term than long-term, change in relatively stable followership behaviors might unfold rather slowly. However, several relationships that we had hypothesized were neither significant in Study 1 nor in Study 2. While our studies still can be an orientation for designing time-lags for followership research, we recommend future research to consider different time lags in order to learn in which time frames the effects of followership behaviors occur and when they potentially decline. Furthermore, future studies could include more than two time points for a more comprehensive understanding of the longitudinal relationships.

Finally, the sample of our first study consisted of employees that were employed within 11 organization from the service sector. The sample of our second study was a heterogeneous convenience sample from an online respondent pool. Hence, we cannot claim generalizability of our findings, even if several findings were consistent across both of our studies. Future research should, however, further investigate followership behavior in various samples and different sectors to study the generalizability of the impact of active and critical followership.

## CONCLUSION

Our study contributes to a better understanding of *Kelley*'s (*1992*) followership behaviors and its relations to important job-related variables (*i.e.,* job attitudes and self-efficacy). Active engagement and independent critical thinking were both found to be stable and rather consistent followership behavior patterns across two different samples and within two different time-lags. Furthermore, we could detect positive relationships between followership behaviors, job satisfaction and organizational commitment, and a negative relationship between independent, critical thinking and self-efficacy above and beyond the autoregressive effects. However, across the two studies, we obtained mixed results for several relationships that we had predicted. While some of our hypotheses were only supported in one of the two studies, other relationships that we had predicted were neither significant in Study 1 nor Study 2. Hence, more research is necessary to explore potential mechanisms (including time) that link followership with relevant outcomes. Still, our studies open up promising avenues for future research and provide starting points for its conceptual designs (*i.e.,* for instance, with regard to the treatment of followership behaviors as rather consistent behavior patterns or the definition of appropriate time-lags for longitudinal followership research).

### Funding
The authors received no funding for this work.

### Competing Interests
The authors declare there are no competing interests.

## Author Contributions

- Mirko Ribbat conceived and designed the studies, performed the studies, analyzed the data, prepared figures and/or tables, authored or reviewed drafts of the article, and approved the final draft.
- Christoph Nohe conceived and designed the studies, authored or reviewed drafts of the article, and approved the final draft.
- Joachim Hüffmeier conceived and designed the studies, authored or reviewed drafts of the article, and approved the final draft.

## Human Ethics

The following information was supplied relating to ethical approvals (*i.e.*, approving body and any reference numbers):

The ethics committee of the University of Münster approved this research (identifier 2022-65-ChN).

## Data Availability

The variables investigated in Study 1 were part of a more comprehensive data set which had been collected for the German Federal Institute for Occupational Safety and Health (BAuA) between 2017 and 2018 (https://www.baua.de/EN/Tasks/Research/Research-projects/f2372.html). The applied data protection concept for this project limits the data access to employees of the BAuA (one author of this article was employed by the BAuA). Interested parties can contact info-zentrum@baua.bund.de to request access.

The Mplus files and the dataset for Study 2 are available at Open Science Framework (OSF):

Ribbat, Mirko. 2023. "Followership Styles Scrutinized: Construct Stability and Relations to Significant Organizational Variables over Time". OSF. September 12. https://osf.io/bz2tn/.

Ribbat, Mirko. 2023. "Followership Styles Scrutinized: Construct Stability and Relations to Significant Organizational Variables over Time (Study 2)". OSF. September 12. https://osf.io/xj64w.

## Supplemental Information

Supplemental information for this article can be found online at http://dx.doi.org/10.7717/peerj.16135#supplemental-information.

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
