# Peer review of "Followership styles scrutinized: temporal consistency and relationships with job attitudes and self-efficacy"

_PeerJ, doi:10.7717/peerj.16135_

## Round 0.1 · original submission · Major Revisions

Dear Dr. Ribbat,

Thank you for submitting your manuscript to PeerJ. Your paper was evaluated by two individuals with very strong expertise in this area. The two reviewers have found your research question and study interesting and timely. At the same time, they have identified several concerns. Based on my independent reading, I agree that your paper has potential for an impactful contribution, but major revisions are necessary. Thus, I am inviting you the option of revising and resubmitting it for consideration.

The reviewers have provided exceptionally thorough and constructive comments. I am very grateful for their contributions. I will not repeat their points but I urge you to attend to all of their points, especially those related to the validity of the findings and your statistical analyses. I also agree with the reviewers the need to improve readability of your manuscript. You are welcome to submit a revision, if you think that you can address all concerns.

Thank you for giving us the opportunity to consider your work, and we wish you all the best with your article.

Yours sincerely,

Andree Hartanto, PhD
Academic Editor
PeerJ

Reviewer 1 ·

Basic reporting

no comment

Experimental design

no comment

Validity of the findings

no comment

Additional comments

The introduction (pages 3 – 5), can be more succinct. This is the hook of the paper, and it could do with a little trimming from p. 5, line 103 onwards. This would also help prevent repetition in the paper. Since it will be repeated right before study 1. The abstract can similarly be more succinct. Authors could consider removing the rhetorical questions in the abstract.

The authors did a good job highlighting the gaps in the current literature on followership behaviors and job-related outcomes (line 83)

Below are some of my questions and comments:
1. If the best followers are those who participate actively and take initiative (line 127), why did the authors predict a positive relationship between AE and job-related outcomes (H2) and a negative relationship between ICT and job-related outcomes (H3)?
a. A related point here is that, it would be better to assess followership behavior as a type? Passive, conformist, alienated, exemplary, pragmatist. It was hard as a reader to reconcile the difference between the theoretical prediction and the actual hypothesis. The hypotheses appear to either be (1) assessing alienated or conformist followership behavior, (2) assessing AE and ICT as independent orthogonal constructs (that they are), but does not align with Kelley’s (1992) theorizing that high AE and high ICT would lead to the best outcomes.
b. The examination of the interaction between AE and ICT predicting job-related outcomes (Hypothesis 8) was insightful and appropriate.
2. The theoretical link between followership and self-efficacy is underdeveloped. Just because it has not been tested is not a good enough reason to test it (line 24 – 242).
a. When followers just take directions, could self-efficacy not arise but successfully fulfilling said directions? It may not contribute to the overall leadership success, but it could just be able not falling behind on my own job/job scope. Akin to a prevention motivation.
b. This may explain the null results found too.
3. Please double check the sample size for Study 1. Is 187 the final sample? Or the overall sample? 187 (line 296) – 3 (line 300) – 5 (line 301) = 179.
4. Please list down the number of items in each scale
5. Please report statistics for non-significant findings too (H6 & H7)
6. Was gender and age correlated with any of the key variables, were there any covariates entered in the analysis?
7. What is AE and ICT’s link with POS? (line 554, hypothesis 14).
8. Does POS and leader humility interact with each other? They are very highly correlated (Table 2, .60 & .63)
9. Was study 1 conducted in German? Or were both studies conducted in German? Please clarify.
10. If followership styles are rather stable behaviour patterns (line 767), this suggests that examination of followership styles as a ‘type’ might be a better idea. However, considering that both AE and ICT are stable, highly correlated, with a high mean, it could be that most participants in both studies belonged to the pragmatist or exemplary style leading to a range restriction. It could be that most participants already adopt the exemplary style and what the results indicate is that there are no relations to job-related outcomes, BUT the outcomes can become dire if participants adopt a conformist or alienated style.
11. The authors could tone down the conclusion that their results are consistent with the assumption that positive effects of followership behavior night depend on organizational culture or shared values… (line 859 – 861).

Minor Issues:
1. For consistency, please split Hypothesis 2 and 3 into Hypothesis 2a, 2b, and Hypothesis 3a, 3b.
2. Grammar: line 227 – 228
3. Any citations for line 259 – 266?
4. Check for APA, italicize all non-Greek symbols (e.g., p. d.) (line 371)
5. Spelling: Forth vs. Fourth (line 684)

Some Suggestion
1. A graph might help reader with the visualization of your hypotheses

Reviewer 2 ·

Basic reporting

1. It should be clear from the beginning what you mean when using the term “follower”. In the followership field, there are two main paradigms to study the topic (the role-based and the process-based). Both have different meanings regarding what it means to be a follower. You clearly build on the role-based approach where followers are equated with subordinates, but you need to also clarify this in your manuscript (See Uhl-Bien et al., 2014 for a review). As it stands, the background of the followership field is a bit missing as well as how your work fits into this field.

2. Your paper would benefit from a visualization of your hypotheses. At the moment, your conceptual model is a bit unclear due to the very high number of hypotheses (15!!!), which makes it sometimes difficult to follow your reasoning behind it.

3. There are a few ambiguous sentences that would benefit from rewriting (e.g., line 189).

Experimental design

1. Your design is flawed and suffers from endogeneity: You measure perceptions of behavior instead of actual behaviors. This is a recurring concern in the leadership literature, namely that scholars are continuing to conflate leader (follower) behaviors with evaluations or perceptions of those behaviors (Banks, Woznyj & Mansfield, In press; Fischer & Sitkin, 2023). You suggest testing theories whereby follower behaviors are exogenous; however, empirically, measures are based on surveys that are affected by a variety of factors such as perceptions, evaluations, values, and behaviors. This is problematic for several reasons.
a. Variables which are measured with scales, perceptions or evaluations of behaviors are endogenous (Antonakis, Bendahan, Jacquart & Lalive, 2010). Endogeneity refers to a situation whereby the error term e correlates with the measured predictor(s) due to unobserved causes of the outcome (e.g., the nature of human cognition to fill blanks and rely on salient perceptual memories allowing unmeasured factors to affect attitudes, cognitions, and perceptions). Moreover, endogeneity is also introduced by omitted variables, reverse causality and simultaneity bias that frequently affect scale measures. When endogeneity is present, the estimated parameter of interest is uninterpretable and this forms a threat to making causal inferences. For instance, you modeled a particular follower behavior (i.e., AE or ICT), x, as a predictor of an outcome like job attitudes or job commitment, y, it is likely that omitted variables at the follower level (e.g., personality), micro level (e.g., department, firm or industry factors), or macro level (e.g., culture) will correlate both with the predictor and the outcome.
b. Potential endogeneity issues should be discussed in the limitations section of the paper, which is currently missing in your paper. There are also several ways to address these issues and you can consider instrumental variable estimation since this was a broader project which measured multiple constructs if I understood correctly. For more information on endogeneity and instrumental variables, see Wulf et al. (2023) as well as Bastardoz et al. (2023).

2. In addition, you rely on a single source of data. You only looked at the self-reports of employees (while you also had access to the leader’s surveys if I understood correctly). This may lead to single source bias and endogeneity, as indicated in the previous point. You state in line 922-926 that this is indeed a limitation in your study, but is there a specific reason why you did not include data from the leaders as well? Otherwise, I would strongly encourage you to consider using data from the leaders as well. Note, even with multi-source data, your model would still suffer from endogeneity and your results could not be interpreted causally.

3. I think your motivation in line 490-502 to use a use a shorter time lag in Study 2 is not sufficiently theorized. If I summarize your thinking, you argue that some effects might have simply vanished over time and that exemplary followership might not always be acknowledged or rewarded. First, I think it is a bit odd that certain effects will vanish over time since you state that AE and ICT are rather stable behavior patterns. Therefore, I do not fully understand how effects may vanish over time then. Moreover, in line 493, you give the example that the respondents’ leader might have changed between the two measurement waves. This argument does not really hold, because if this is the case, then you should perform sensitivity analyses with and without these observations to see how sensitive your results are to including these observations. Second, your argumentation in line 494-502 is generally limited and would need to be better theorized. For instance, I do not understand your argument that exemplary followership might not always be acknowledged or rewarded. How does this change when you shorten the time-lag from 9 to 4 months? Because in my opinion, the missing acknowledgement of follower’s effort should not have a different effect on AE in 4 or 9 to 12 months. In general, I think your reasoning for using a shorter time-lag in Study 2 needs more theorizing.

Validity of the findings

1. I am not convinced by your general contributions and especially the practical implications it entails. One main reason is that you stimulate the use of “Identify Your Followership Style Questionnaire” by Kelley (1992) to assess employees own “behavioral” tendencies. This could maybe work in a hierarchical organization where the leader is equated with a manager/supervisor and the follower with a subordinate. However, a lot of organizations are starting to invest in other organizational structures where the leader/follower role can shift depending on certain situations and contexts (e.g., shared leadership teams, self-organizing teams, or agile teams). Moreover, I don’t see the added value of putting people in boxes according to their follower style, especially when used in the hiring process. A more critical reflection on the practical implications is likely necessary.

2. You conclude in line 766-775 that AE and ICT are rather stable behavior patterns instead of more state like. Then, you argue that a less dynamic perspective on followership behaviors is also appropriate to study followership instead of a dynamic perspective of leader and follower behaviors. It seems that you did not provide much effort to link your work and findings to existing knowledge, as well as how your findings provide more clarity in the different ongoing conversations in the field. For instance, you do not really measure real behaviors, but evaluations and perceptions of behaviors and you also do it very statically via scale measurements in a questionnaire. Therefore, your argument in line 766-775 lacks support and should be adjusted. You can argue that this is indeed in line with the role-based approach where they look at the right mix of follower characteristics and follower behavior to promote desired outcomes. However, the process approach takes a whole new perspective where they look at the followership as part of a dynamic relational process. Here, they use totally different methods to study these dynamic relations so it is difficult to make a comparison between the two as you did. So I would encourage you to entrench the findings of your study in the context of the followership field.

Additional comments

Thank you very much for offering me the opportunity to review your work. I see the relevance of the research topic and appreciate and encourage studies testing the relationship between followership behaviors and important job outcomes. The interest in followership as an important part in the leadership process is growing in today’s work environment. However, I see important issues that would need to be taken care.

I wish you all the best as you continue this line of research.

References
• Antonakis, J., Bendahan, S., Jacquart, P., & Lalive, R. (2010). On making causal claims: A review and recommendations. The Leadership Quarterly, 21(6), 1086-1120.
• Banks, G. C., Woznyj, H. M., & Mansfield, C. A. (2021). Where is “behavior” in organizational behavior? A call for a revolution in leadership research and beyond. The Leadership Quarterly, 101581.
• Bastardoz, N., Matthews, M. J., Sajons, G. B., Ransom, T., Kelemen, T. K., & Matthews, S. H. (2023). Instrumental variables estimation: Assumptions, pitfalls, and guidelines. The Leadership Quarterly, 34(1), 101673.
• Fischer, T., & Sitkin, S. B. (2023). Leadership styles: a comprehensive assessment and way forward. Academy of Management Annals, 17(1), 331-372.
• Uhl-Bien, M., Riggio, R. E., Lowe, K. B., & Carsten, M. K. (2014). Followership theory: A review and research agenda. The Leadership Quarterly, 25(1), 83-104.
• Wulff, J. N., Sajons, G. B., Pogrebna, G., Lonati, S., Bastardoz, N., Banks, G. C., & Antonakis, J. (2023). Common methodological mistakes. The Leadership Quarterly, 34(1), 101677.

Annotated reviews are not available for download in order to protect the identity of reviewers who chose to remain anonymous.

---

## Round 0.2 · accepted · Accept

Dear Dr. Ribbat,

Thank you for your diligent efforts in addressing the concerns raised by the reviewers. Although I attempted to send your revised manuscript to our reviewers for further comments, I did not receive any responses. Nevertheless, upon my independent reading and evaluation, I agree that the current version of the manuscript has significantly improved.

I appreciate your thoroughness in addressing the reviewers' concerns. Consequently, I am pleased to advise that the above paper has now been accepted for publication in PeerJ. Thank you for giving the Journal the opportunity to publish your work.

Best Regards,
Andree